# High-strength and crack-free welding of 2024 aluminium alloy via Zr-core-Al-shell wire

Jun Jin[1], Shaoning Geng [1]✉, Leshi Shu[1], Ping Jiang [1]✉, Xinyu Shao[1], Chu Han[1], Liangyuan Ren[1], Yuantai Li[1], Lu Yang[1] & Xiangqi Wang[2]

The 2000 series aluminium alloys are qualified for widespread use in lightweight structures, but solidification cracking during fusion welding has been a long-standing issue. Here, we create a zirconium (Zr)-core-aluminium (Al)-shell wire (ZCASW) and employ the oscillating laser-arc hybrid welding technique to control solidification during welding, and ultimately achieve reliable and crack-free welding of 2024 aluminium alloy. We select Zr wires with an ideal lattice match to Al based on crystallographic information and wind them by the Al wires with similar chemical components to the parent material. Crack-free, equiaxed (where the length, width and height of the grains are roughly equal), fine-grained microstructures are acquired, thereby considerably increasing the tensile strength over that of conventional fusion welding joints, and even comparable to that of friction stir welding joints. This work has important engineering application value in welding of high-strength aluminum alloys.

Today, lightweight materials are an important component in promoting energy and environmental sustainabiliy[1,2]. Every additional 100 kg decrease in vehicle weight leads to a reduction in $CO_2$ emissions of 8.7 g per km and fuel consumption of 0.4 l per 100 km[3]. Therefore, the assembly of lightweight structural parts into a functional unit holds great significance across multiple industries, including aerospace, railway, automotive and shipbuilding[4]. Welding is an important process for assembling lightweight materials.

Aluminium (Al) alloys are typical lightweight materials and have found extensive applications in recent decades[5,6]. Owing to their exceptional strength at low weight[7,8], the 2xxx series alloys are often employed in aerospace and military field. One of the most widely studied members is Al alloy 2024 (AA2024). For the welding of AA2024, friction stir welding (FSW) is favored by most experts[9]. However, the FSW process tends to be constrained by complicated shape of structural parts and requires special set-up. While fusion welding is undoubtedly more flexible and efficient, numerous scholars have conducted research on the welding of AA2024, such as arc welding[10], laser welding[11], electron beam welding[12] and hybrid welding[8]. And they highlight that one of the primary problems of

fusion welding is solidification cracking, which considerably hinders its widespread use.

AA2024 tends to have a relatively wide temperature range that leads to wide mushy zone where liquid and solid phases coexist during solidification, and often solidifies in the form of columnar dendrites during the fusion welding[13]. As the solidification process advances, the proportion of liquid phase decreases. When the tensile stress resulting from solidification shrinkage exceeds the strength of the almost completely solidified microstructure and the liquid feeding is insufficient during solidification, solidification cracking would occur between the dendrites[13].

To expand the potential use of AA2024 in aerospace and military applications, many efforts have been made to address the solidification cracking issues[14–16]. The development of filler materials provides effective methods for inhibiting solidification cracking[8,17,18]. As we know, the filler material is usually similar to the base metal to preserve its original properties without compatibility problems. Based on this principle, ER2319 filler material, an Al-Cu wire, should be the priority selection for welding AA2024. However, this strategy is not suggested due to the occurrence of solidification cracking. Instead, ER4043 or

[1]The State Key Laboratory of Intelligent Manufacturing Equipment and Technology, School of Mechanical Science and Engineering, Huazhong University of Science & Technology, Wuhan, Hubei 430074, PR China. [2]Jihua Laboratory Testing Center, Ji Hua Laboratory, Foshan, PR China. ✉e-mail: sngeng@163.com; jiangping@hust.edu.cn

ER4047 filler materials—a binary Al-Si system—are widely suggested to join these materials[13,17]. This is because the introduction of Si element could promote to form a substantial quantity of low-melting-point eutectic, which is able to rapidly fill the channels between the dendrites and cure the cracking through the healing effect[13]. It should be noted that Al-Si systems filler material can mitigate solidification cracking, but the weld joints generally exhibit unsatisfactory mechanical properties. For example, the tensile strength of weld joints fabricated with Al-Si wires converges at approximately 280 MPa[8], while the strength of AA2024 used in the aerospace community exceeds 400 MPa. Thus, achieving high-strength and crack-free welding of AA2024 remains a barrier.

Delightfully, fine equiaxed microstructures exhibit superior strain accommodation capabilities in the mushy zone by mitigating coherency that constrains the dendrites orientation and facilitates cracking[19]. Meanwhile, fine equiaxed microstructures also present good performance[15], which has been proven by numerous scholars. Nevertheless, how to obtain the fine equiaxed microstructure has been a primary problem. Promisingly, introducing nucleation particles to produce identical ultrafine equiaxed structure has been an effective method. It can enlarge the equiaxed region of the thermal gradient-growth velocity curve[20], which could easily assist to generate equiaxed microstructure[21]. Furthermore, the emergence of nucleation particles could increase the undercooling at the solid/liquid interface and decrease the critical nuclear radius[22], thereby effectively facilitating the grain refinement during solidification. To obtain ultrafine equiaxed microstructures, the nucleation particles need to have similar lattice parameters to $\alpha$-Al[22–24]. Thus, common elements for inoculation treatments, such as Zr, Ti, and Sc, have been chosen to form $Al_3X$ ($X$ = Zr, Ti, or Sc) owing to their small lattice mismatch with $\alpha$-Al. The lattice parameter of $Al_3Zr$, $Al_3Sc$ and $Al_3Ti$ is 4.08 Å, 4.103 Å and 3.967 Å, respectively, which is similar to that of Al of 4.049 Å[23]. Compared to $Al_3Sc$ or $Al_3Ti$ phase, the lower misfit value (0.765%) between the $Al_3Zr$ phase and $\alpha$-Al phase can decrease the nucleation barrier for precipitation[24]. Moreover, the $Al_3Zr$ phase can serve as excellent heterogeneous nucleation sites of $\alpha$-Al due to the close structural resemblances between the two phases[25]. For example, two kinds of cube-on-cube orientation relationships (OR) exist between $L1_2$-$Al_3Zr$ and $\alpha$-Al: Al(001)//$L1_2$-$Al_3Zr$(001), Al[110]//$L1_2$-$Al_3Zr$[110] and Al(100)//$L1_2$-$Al_3Zr$(100), Al[010]//$L1_2$-$Al_3Zr$[010][26].

In this work, we create a zirconium (Zr)-core-aluminium (Al)-shell wire (ZCASW) and employ the oscillating laser-arc hybrid welding technique to control solidification during welding, and fine grain (approximately 4 μm), crack-free of AA2024 weld joints are obtained. The welds display a surprisingly high ultimate tensile strength of 349 MPa that is even comparable to that of FSW. This suggests that the introduction of the ZCASW could enable fusion welding of AA2024 for structural designs in the aerospace industry. The filler material overcomes the dilemma of the trade-off between strength and cracking that remains an open question in applications of high-performance alloys. This welding technology provides a foundation for broad industrial applications because it could meet the demands for efficiency in automated welding.

## Results

### Macroscopic appearance of welded joints

Figure 1a illustrates the oscillating laser-arc hybrid welding process of AA2024, where the filler material is melted by the arc heat source, and the oscillating laser beam is applied behind the arc to promote the alloy element distribute homogeneously. The innovative ZCASW is fabricated by assembling 7 fine wires (diameters of 0.6 mm). The Zr wire serves as the central wire, around which six Al wires are twisted and wound (see Methods). Figure 1b displays the schematic diagram and product of the ZCASW filler material. A key benefit of this kind of configuration lies in how the grain refiner elements are incorporated

into the molten pool. During welding, the first metallurgical bond among elements occurs with the formation of droplet, and then each droplet becomes increasingly larger until it enters the molten pool, where it could be mixed and stirred to obtain the second metallurgical bond driven by the oscillating beam. This leads to a more effective and homogeneous transition of alloying elements. Furthermore, a simple approach for the preparation of the ZCASW filler material solely relies on the physical strand mechanism instead of conventional casting process, which has shorter preparation times and less cost of critical raw materials to some extent. In terms of use, we also demonstrate that the ZCASW filler material meets the efficiency demands for automated welding, just like conventional welding wire. Therefore, we believe that this welding wire has significant promise for use in industrial manufacturing. To obtain fine equiaxed microstructures to resist solidification cracking, Zr is selected as the grain refiner, which results in the formation of a favorable $Al_3Zr$ nucleation phase. This phase could significantly improve grain refinement efficiency and provide a low-energy nucleation barrier for $\alpha$-Al according to classical nucleation theory[22,24]. Figure 1c displays the crystallographic data of $\alpha$-Al and $L1_2$-$Al_3Zr$ and their lattice match relationships[15,26,27].

We perform laser welding experiments using the ZCASW filler to fuse AA2024 sheets with dimensions of 150 mm × 100 mm × 8 mm. Meanwhile, we perform control experiments using standard ER2319 (Al-Cu) filler and ER4043 (Al-Si) filler. Figure 2a shows the surface topography of the welding seam fabricated with the ZCASW filler and perfect formation is obtained. Figure 2b and Fig. 2c demonstrate the longitudinal and cross section morphology of the welding seam, respectively. No cracks can be observed under the optical microscope. To visualize the weld surface morphology, we employ a 3D depth-of-field optical microscope to measure the height of the top surface of welding seam that presents favorable uniformity, as shown in Fig. 2d. To further visualize the 3D nature of internal imperfections, a 2 mm × 2 mm × 5 mm microvolume is excised for examination by X-ray micro computed tomography technology. Figure 2e, f, g show the 3D-reconstruction volumes of the defects inside the welding seam using the ER2319 filler, the ER4043 filler and the ZCASW filler, respectively. Of notes, the cracks (rendered in yellow) appear in the Fig. 2e. Figure 2f displays the anticipated effect and no crack appears. Figure 2g exhibits no cracks as well, indicating that this filler material is remarkably effective.

### Modification of microscopic structure

The shape of the solid primary dendrites/crystals is known to influence cracking susceptibility during solidification. In general, solidified microstructures with coarse dendrites have higher solidification cracking susceptibility (SCS) than those with fine dendrites[13,15]. To further validate these results, we conduct microstructure analysis from the perspective of the dendrite morphology and the interdendritic region morphology in the melting zones fabricated with different fillers.

Here, we first discuss the effect of the dendrite morphology in detail. Figure 3a presents the melting zone fabricated with the ER2319 filler material. Owing to the wide mushy zone and the nonlinear relationship between the solid fraction and temperature of the high-strength Al alloy[4], the melting zone solidifies into large dendrites. Figure 3b shows the inverse pole figure (IPF) of the melting zone. To quantify the dendrite size, we count and obtain an average size of 36 μm from the melting zone (Fig. 3c). It should be noted that large dendrites could lead to the long interdendritic liquid channels during solidification. As the temperature and liquid volume fraction decrease, the long liquid channels could be trapped or hindered by the developing dendritic solid network. Solidification cracking would occur when liquid feeding becomes insufficient to compensate for solidification shrinkage and thermal contraction[28,29]. Figure 3d displays the melting zone fabricated with the ER4043 filler material. The average

dendrite size in the melting zone is 38 μm (Fig. 3f), which is close to that of the ER2319 filler (36 μm). No solidification cracking is found in the melting zone. However, several compromises must be made in the employment of this method. The molten filler alloy locally dilutes the molten AA2024 base alloy, which reduces the concentration of strengthening alloy[4]. This generally leads to the reduction in the strength of welded joints.

The use of the ZCASW filler material successfully alters the solidification mechanism during welding of AA2024. The welds are free of cracks under the premise of ensuring good formation. Figure 3g shows the homogeneous microstructure throughout the melting zone fabricated with the ZCASW filler, which is remarkably distinct from those obtained with ER2319 and ER4043 filler materials. The dendrites exhibit highly equiaxed morphology with an average size of 4.0 μm (Fig. 3i), and

the interdendritic regions are smooth. In general, the molten pool is dominated by large dendrites during welding due to the primary solidification condition favouring epitaxial growth, unless vigorous nucleation events occur ahead of the solid/liquid front[30]. StJohn et al.[31] reported that equiaxed and fine microstructures were favored by potent nucleation particles at high number densities as well as large values of the growth-restricting factor (Q), which can be expressed as $Q = m_l C_0 (k-1)$[30]. Here, $m_l$ represents the gradient of liquidus, $C_O$ represents the initial composition of alloy, and $k$ represents the partition coefficient. The introduction of Zr not only increases the Q value but also provides potent nucleation particles of $Al_3Zr$[30]. Therefore, the larger grain-restricting factor and more potent nucleants could contribute to the highly equiaxed and fine microstructure in our welds (Fig. 3g). To further reveal the growth behavior of highly equiaxed dendrites owing to

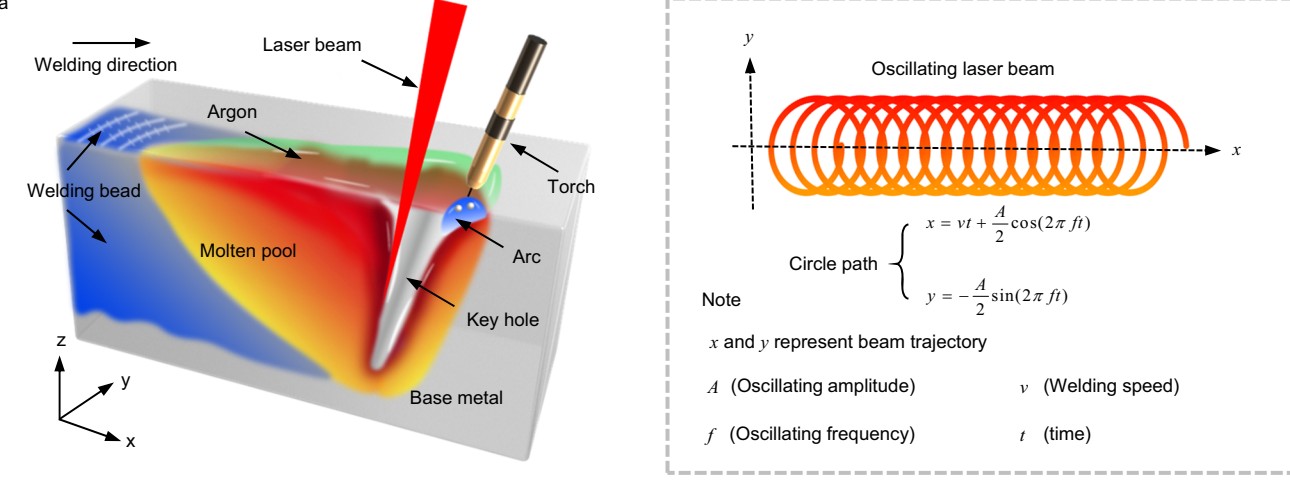

Hybrid laser-arc welding with oscillating mode

$$x = vt + \frac{A}{2}\cos(2\pi ft)$$
$$y = -\frac{A}{2}\sin(2\pi ft)$$

Note: $x$ and $y$ represent beam trajectory

$A$ (Oscillating amplitude)   $v$ (Welding speed)

$f$ (Oscillating frequency)   $t$ (time)

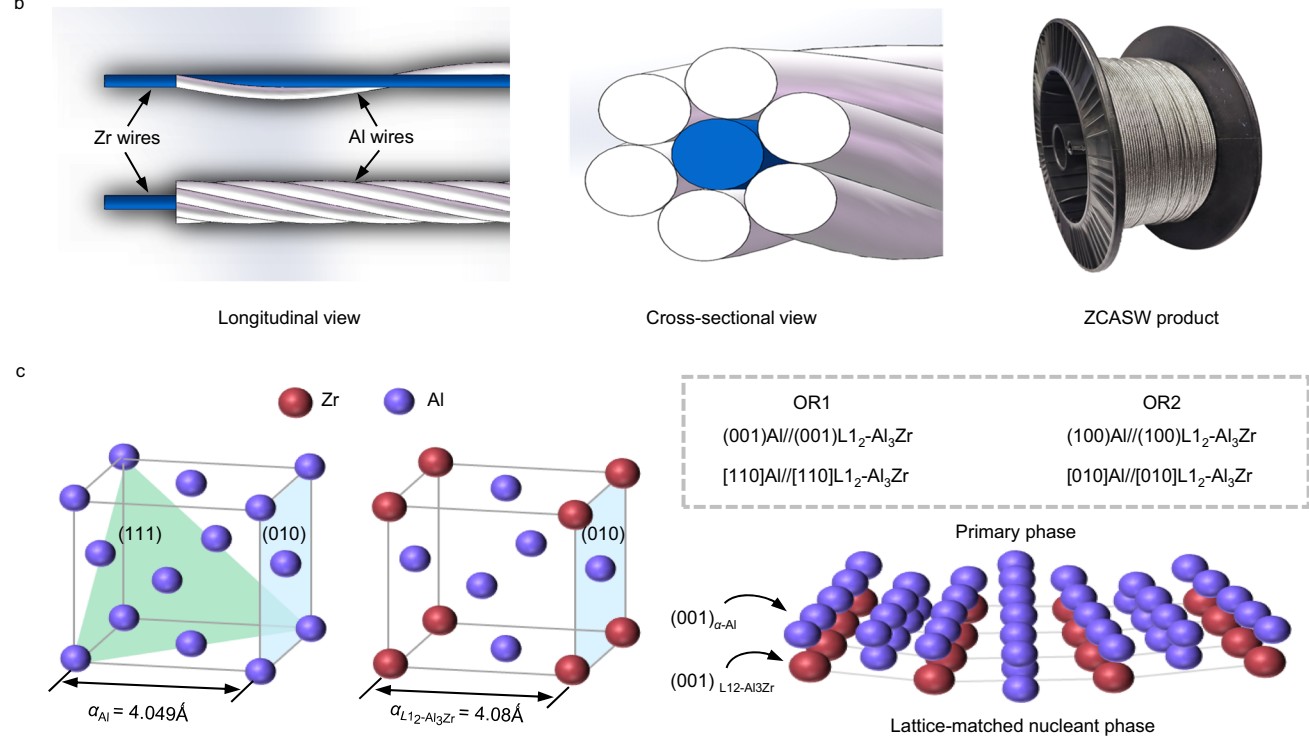

**Fig. 1 | Welding process of AA2024 and the ZCASW filler material. a** Schematic diagram of the welding process. **b** Longitudinal and cross-sectional schematic diagram and product of the ZCASW filler. **c** Schematic representation of the crystallographic data of α-Al and $L1_2$-$Al_3Zr$[15,26,27], illustrating how the lattice-matched $Al_3Zr$ phase could induce low-energy-barrier epitaxial growth of α-Al (OR1 represents orientation relationship 1 and OR2 represents orientation relationship 2).

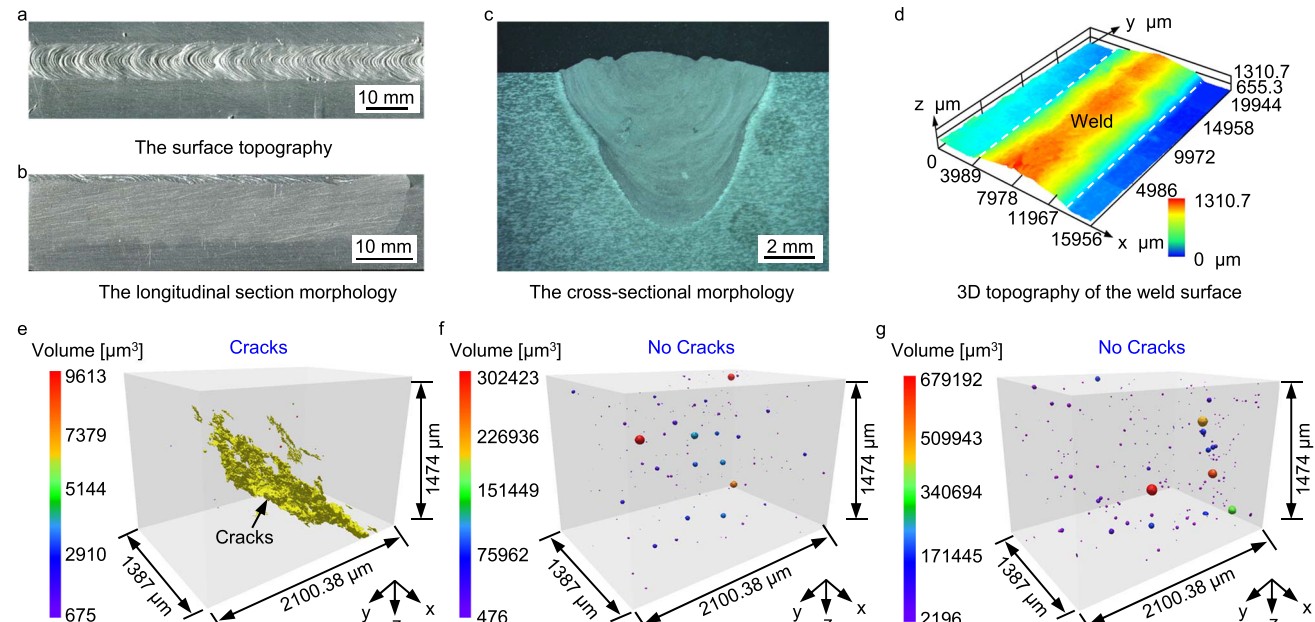

**Fig. 2 | The macroscopic morphology and microscopic defects of the welding seam. a** The surface topography of the welding seam fabricated with ZCASW filler. **b** The longitudinal section morphology of the welding seam. **c** The cross-sectional morphology of the welding seam. **d** The morphology and height measurements of the weld surface (Color bar represents the height of the surface of welding seam). **e–g** 3D-reconstruction volumes of the defects inside the welding seam fabricated with the ER2319, the ER4043 and the ZCASW fillers, respectively (Color bar denotes pores diameter).

the addition of Zr, typical equiaxed regions with intragranular precipitates are excised by a Focused ion beam (FIB) for Transmission electron microscopy (TEM) analysis. Figure 3j shows a Scanning TEM image acquired in High angular annular dark field (STEM-HAADF) mode in one equiaxed dendrites region. Additionally, the precipitate is very rich in Zr and Al according to the Energy dispersive X-ray spectroscopy (EDX) mapping image (Fig. 3k). To further identify the crystallographic structure of Zr-rich precipitates in the equiaxed dendrites region, Selected area electron diffraction (SAED) is performed. The diffraction spots associated with the Zr precipitates are circled in yellow on the SAED pattern (Fig. 3l). Diffraction patterns for several Zr cubic precipitates reveal an $L1_2$ faced-centered cubic structure (FCC), supporting the formation of $L1_2$-$Al_3Zr$ precipitates. Figure 3m shows SAED patterns of the Al matrix in the [01(_)1] direction, and the diffraction spots are marked in yellow. The diffraction spots associated with $L1_2$-$Al_3Zr$ are marked in blue. An orientation relationship (OR) between $L1_2$-$Al_3Zr$ and $\alpha$-Al is observed following $[01(_)1]_{Al}//[01(_)1]_{L12-Al3Zr}$. This OR demonstrates the typical cube-on-cube relationship between these two phases[25]. Figure 3n shows the high-resolution STEM-HAADF images taken at an $\alpha$-Al/$L1_2$-$Al_3Zr$ interface, and the incident electron beam is parallel to the [01(_)1] zone axis of $L1_2$-$Al_3Zr$ (upper part) and the adjacent Al grain (bottom part). The corresponding Fast Fourier transformation (FFT) pattern is displayed by the insert in Fig. 3n. The interfacial atomic coincidence can be clearly seen in Fig. 3o, which demonstrates a fully coherent interface between these two crystals. Such full coherency is attributed to the smaller interplanar mismatch between these two phases. The planar distances of (200) $Al_3Zr$ and (200) $\alpha$-Al are 0.2084 nm and 0.2064 nm, respectively. Thus, the misfit at the $Al_3Zr$-$\alpha$-Al interface is calculated to be approximately 0.96%, indicating a highly coherent interface that provides provide a low-energy nucleation barrier for $\alpha$-Al[32,33].

The next point to consider is the variation in the interdendritic phase morphology. Scanning electron microscope (SEM) is employed to investigate the interdendritic phase morphology in the melting zone fabricated with the ER2319 and the ZCASW filler materials. Figure 4a illustrates the typical interdendritic phase distribution in the melting zone fabricated with the ER2319 filler. The interdendritic phase is lamellar and continuously distributes along the interdendritic regions, normally resulting in continuous segregations. This dendrite growth is unexpected since it generally causes the melting zone to have less tensile strength and trigger crack defects even if there is less stress or strain[13].

Figure 4b displays the typical interdendritic phase distribution in the melting zone fabricated with the ZCASW filler. Here, the interdendritic phase is segmented, and its fragments are randomly oriented, shorter than their counterparts fabricated with the ER2319 filler. Meanwhile, the area fractions of interdendrictic phase obtained from different melting zones are further analyzed by SEM images in the Backscattered electron (BSE) diffraction mode. The area fraction of the interdendritic phase in the melting zone fabricated with ER2319 filler is 11.5% (Fig. 4c), and it increases to 15.95% in the melting zone fabricated with ZCASW filler. More interdendritic phase indicates more liquid exists among dendrites during the terminal stage of solidification[34], which could sufficiently compensate for solidification shrinkage and thermal contraction. On the other hand, the shorter and randomly oriented interdendritic phase, combined with highly equiaxed and round dendrites, indicate high fluidity of the remaining melt during solidification. The current results are compatible with the less susceptibility to cracking of the melting zone.

To further clearly identify the composition distribution of melting zone, a typical region is selected for Electron probe micro analysis (EPMA) analysis (Supplementary Fig. 6). Figure 4d shows the representative results. The interdendritic phase is highly enriched in the Mg and Cu elements, and Zr dominates the interiors of dendrites. To further identify the crystallographic information of the interdendritic phase, a typical region is excised by FIB for TEM analysis (Fig. 4e). A typical intergranular phase region is placed in STEM-HAADF mode and identified in combination with SAED and EDX mapping analyses. Figure 4f, g show the representative results, which are consistent with the $Mg_2Cu$ phase. Figure 4h shows a HRTEM image of the interface between the $\alpha$-Al and $Mg_2Cu$ phase, and the inset illustrates the FFT of the $Mg_2Cu$ phase.

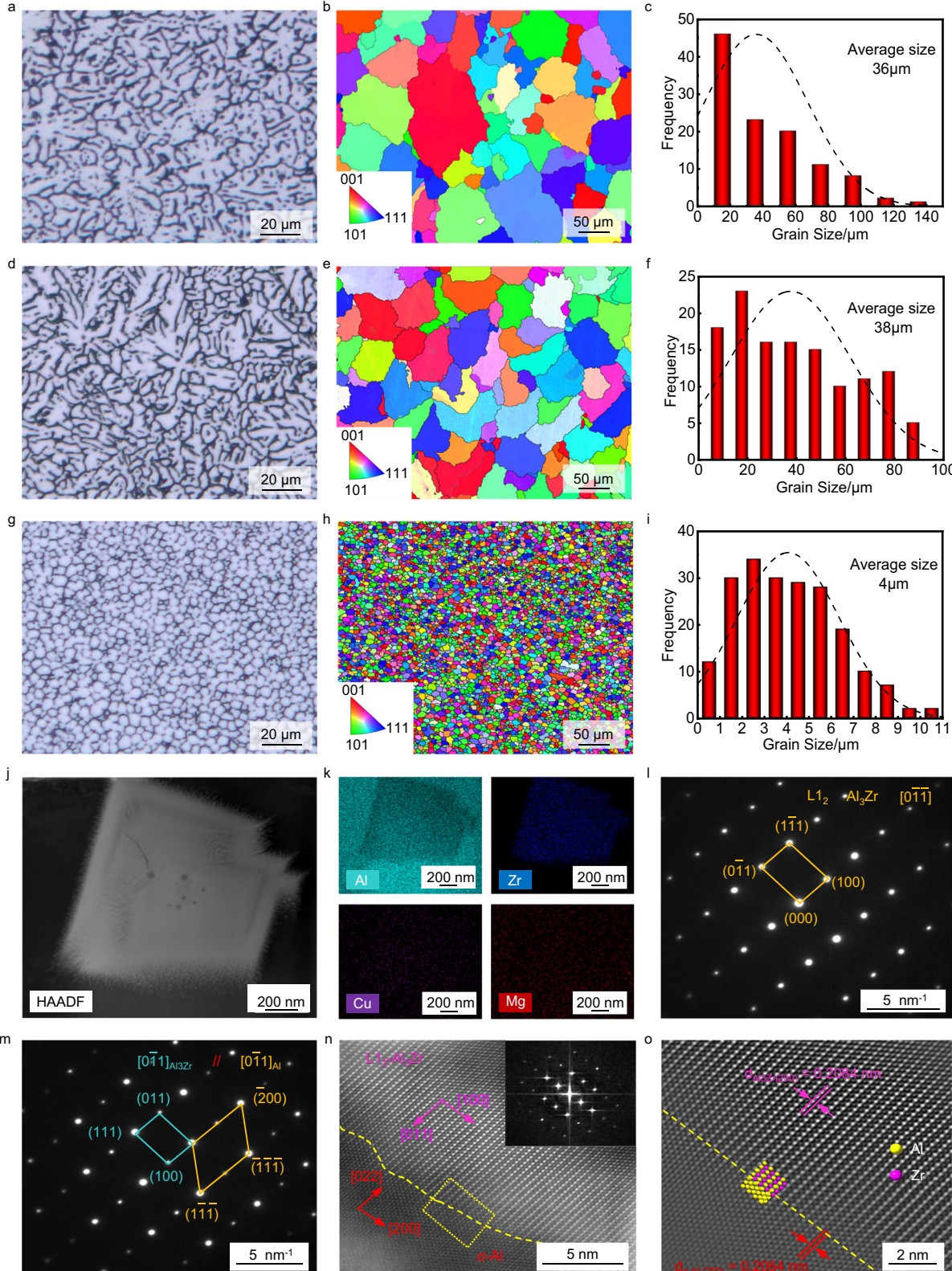

**Fig. 3 | Microstructural features of weld cross-sections fabricated with different filler materials. a**, **d**, **g** Ultra-depth-of-field microscope characterization of welds fabricated with the ER2319, the ER4043 and the ZCASW filler materials, respectively. **b**, **e**, **h** The IPF of the melting zones fabricated with the ER2319, the ER4043 and the ZCASW filler materials. **c**, **f**, **i** Dendrites size statistics of the melting zones fabricated with the ER2319, the ER4043 and the ZCASW filler materials. **j** STEM-HAADF image in one equiaxed dendrites region and corresponding EDX mapping of the main elements (Al, Zr, Cu and Mg) shown in (**k**). **l** Diffraction pattern of the L1$_2$-Al$_3$Zr precipitate in the [01($_$)1($_$)] direction. **m** Diffraction pattern of the interface between the Al matrix and L1$_2$-Al$_3$Zr in the [01($_$)1] direction. **n** HRSTEM-HAADF image taken at the α-Al/L1$_2$-Al$_3$Zr interface along the [01($_$)1]$_{Al}$([01($_$)1]$_{L12-Al3Zr}$) zone axis, with the inset showing the corresponding FFT pattern. **o** Magnified view of the marked yellow area in (**n**), showing atomic coincidence across the interface between these two crystals.

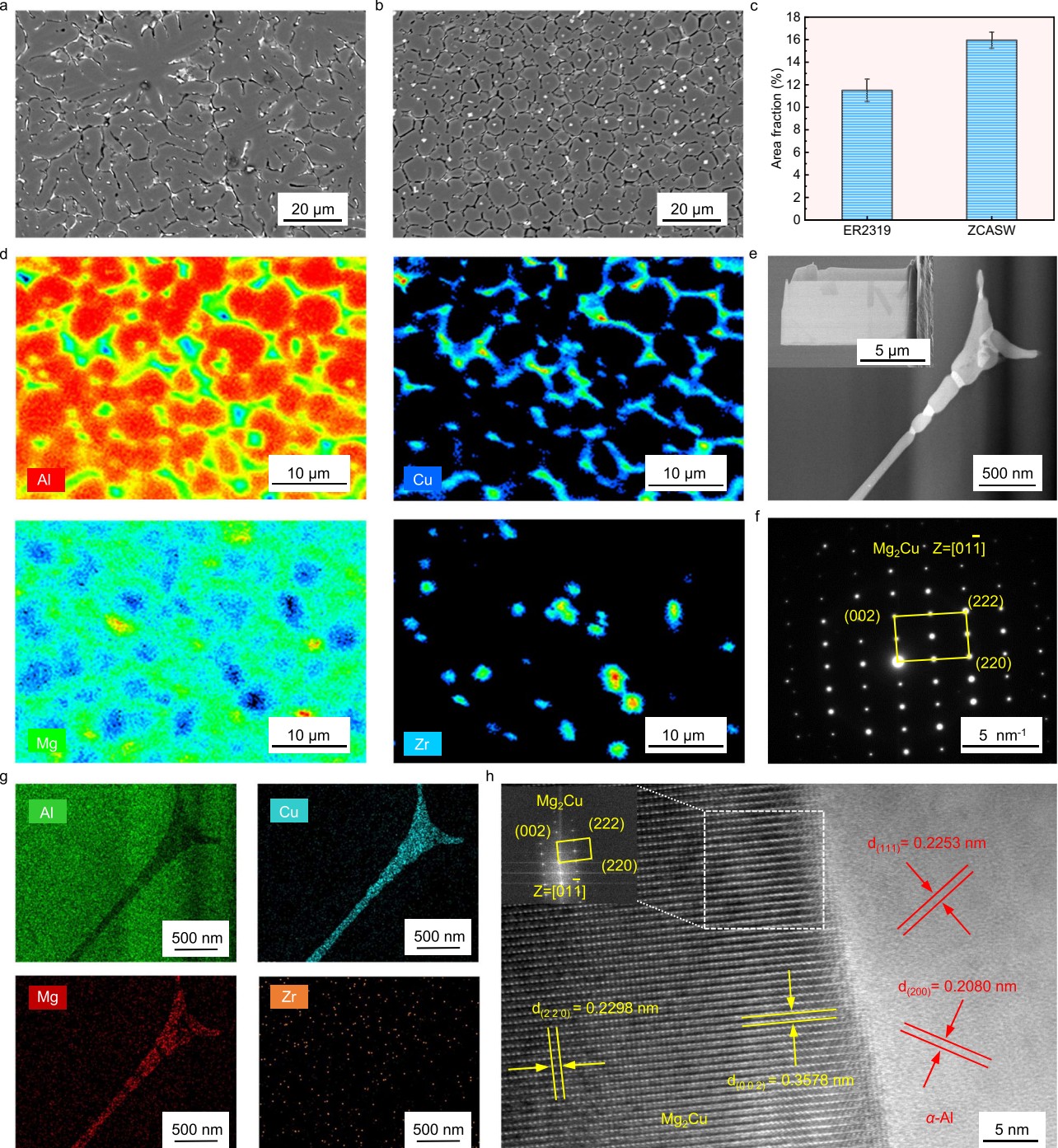

**Fig. 4 | Interdendritic phase morphology characterization in the melting zone fabricated with the ER2319 and the ZCASW filler materials. a**, **b** Interdendritic phase in the melting zone fabricated with the ER2319 and the ZCASW filler materials. **c** Comparison of interdendrictic phase area fractions obtained from different melting zones (Error bars represent standard deviation). **d** EPMA maps showing the chemical constitutions of the interdendritic phase and Al matrix in the melting zone fabricated with the ZCASW filler. **e** Interdendritic phase observed by TEM-EDX, showing a STEM-HAADF image. **f** SAED pattern of the interdendritic phase. **g** EDX mapping of the main elements (Al, Cu, Mg and Zr). **h** HRTEM image of the interface between $\alpha$-Al and $Mg_2Cu$ phase, with the inset showing the FFT pattern of $Mg_2Cu$ phase.

## Mechanisms of solidification cracking inhibition

To further reveal how the grain refinement affects solidification cracking, we propose an approach combining the phase field (PF) simulation and Kou's criterion for cracking. The PF simulation is undertaken to investigate the microstructural evolution that significantly influences solidification cracking. The Kou's criterion for cracking[35], which primarily considers grain separation owing to tensile deformation, the growth rate of grains to bond together, and liquid feeding to fill grain boundaries, is employed to investigate the SCS. In this criterion, the maximum $|dT/d(f_s)^{1/2}|$ of an alloy, where $T$ refers to temperature and $f_s$ represents the solid fraction, is proposed as a simple index for assessing the SCS[35]. This index is related to the $T$-$(f_s)^{1/2}$ curve, and a more precise $T$-$(f_s)^{1/2}$ relationship could more accurately predict the SCS.

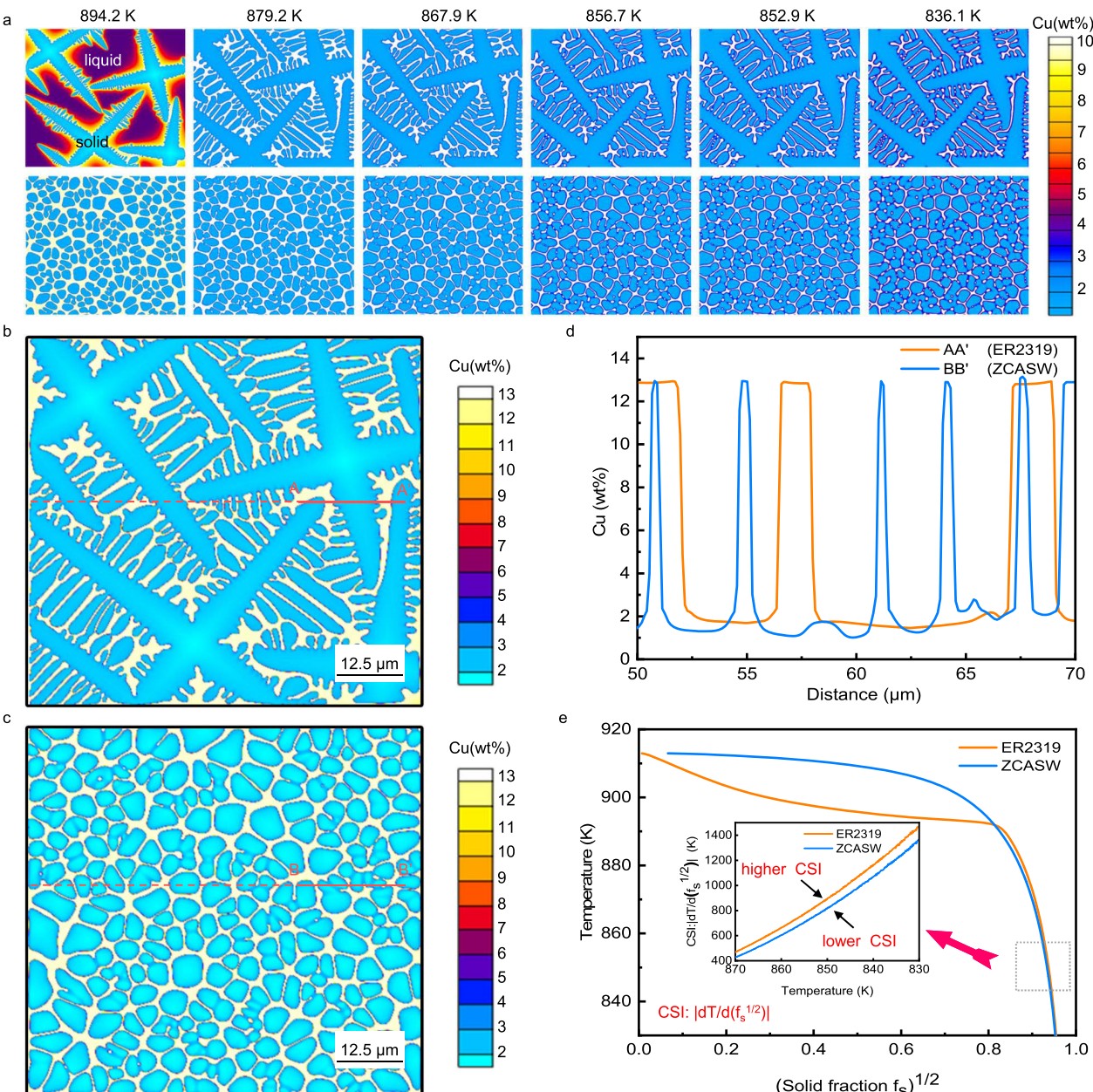

**Fig. 5 | Solidification behavior of the melting zone fabricated with the ER2319 and the ZCASW fillers. a** Simulated liquid channel morphology during dendritic growth of the melting zone fabricated with ER2319 (top) and ZCASW (bottom) fillers. The six panels in each row correspond to the different temperatures. **b**, **c** Simulated solute distribution of the melting zone fabricated with the ER2319

filler and the ZCASW filler at 882 K. **d** Solute distribution along line AA' in (**b**) and line BB' in (**c**). **e** $T$-$(f_s)^{1/2}$ curves and cracking susceptibility indexes calculated from the PF simulations for the melting zone fabricated with the ZCASW (blue) and the ER2319 (orange) fillers (CSI represents cracking susceptibility index).

Figure 5a simulates the variation in dendrites and liquid channel morphologies during solidification of the melting zone fabricated with the ER2319 (top) and the ZCASW (bottom) fillers. When the average dendrite size is 36 μm, corresponding to ER2319 filler, the microstructure exhibits dendritic growth and the well-developed side branches are accompanied with the primary arms. When the dendrite size decreases to 4.0 μm, corresponding to the ZCASW filler, the microstructure appears to be a nondendritic equiaxed structure, and no side branches formed. These simulated results are consistent with the previous experimental results. An additional point of concern is the liquid channel morphologies at the terminal solidification stage. As for the ER2319 filler, the residual liquid firstly forms continuous elongated liquid channels, and then the liquid channels become undulant due to the appearance of side branches. However, the liquid channels appear

significantly shorter between dendrites as for ZCASW filler, and their solidification path is often more tortuous, which makes a great contribution to resisting cracking. In addition, Fig. 5a also illustrates the solute distribution in the melting zones fabricated with the ER2319 and the ZCASW fillers. During dendritic growth, the solute is rejected to the residual liquid by the solid network structure. As the temperature decreases, the concentration of solute Cu is much higher in the liquid. As for the ER2319 filler, it is clearly that solute segregation emerges not only between the primary dendrites but also between the secondary dendrites. However, the solute segregation behavior only occurs between nondendritic equiaxed structure as for the ZCASW filler.

To better understand the solute segregation behavior, the Cu concentration distribution along line AA' in Fig. 5b and line BB' in Fig. 5c is shown in Fig. 5d. Note that the solute distribution maps are

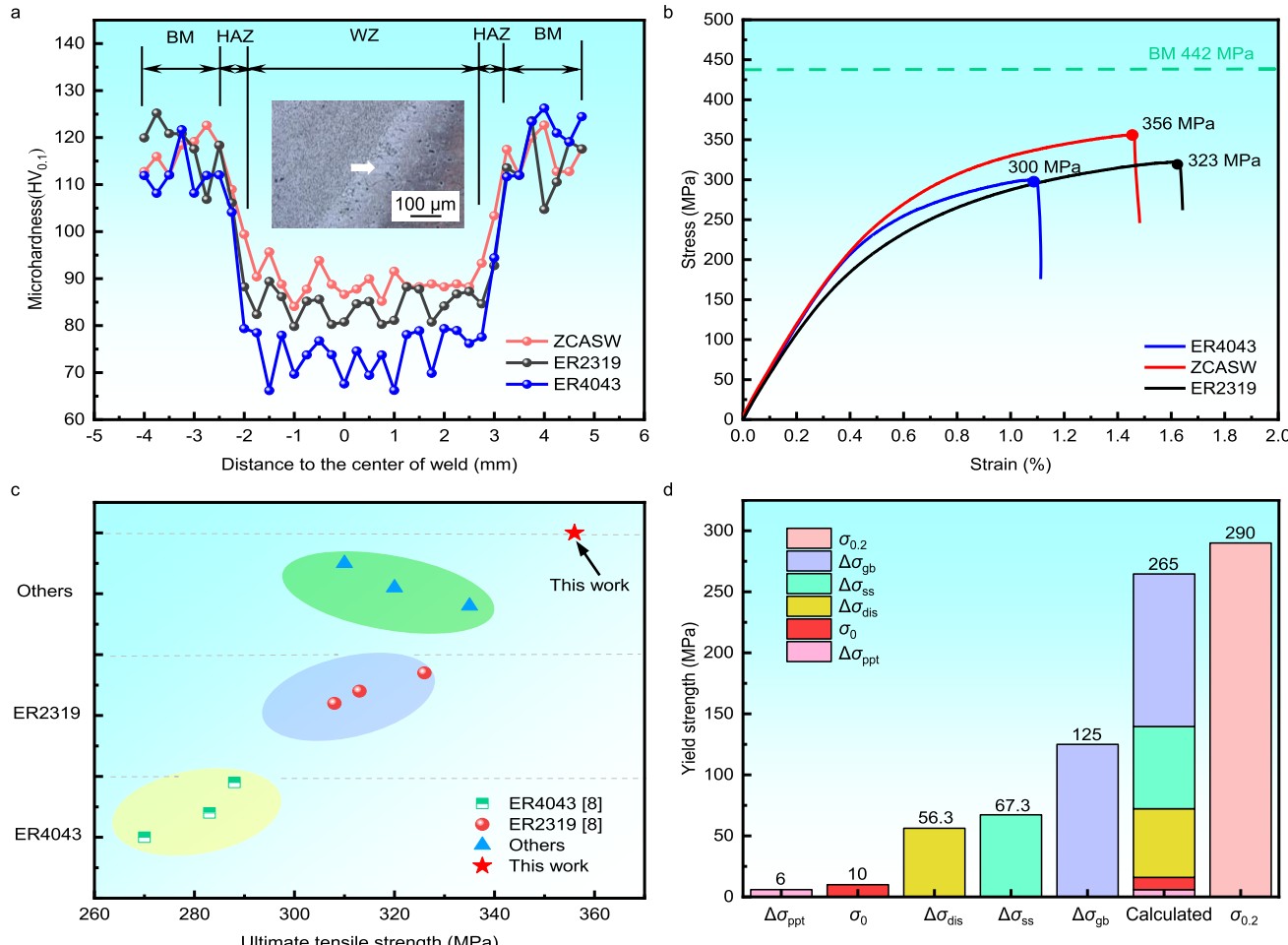

**Fig. 6 | Mechanical properties of welds fabricated with the ER2319, the ER4043 and the ZCASW fillers. a** The microhardness of welds fabricated with the ER2319, the ER4043 and the ZCASW fillers. **b** Engineering stress-strain curves of the welds fabricated with the ER2319[8], the ER4043[8] and the ZCASW fillers. **c** A summary of the tensile strength for the welds fabricated with various filler materials.

**d** The estimated strengthening contributions from the precipitation strengthening ($\Delta\sigma_{ppt}$), grain size strengthening ($\Delta\sigma_{gb}$), solid solution strengthening ($\Delta\sigma_{ss}$) and dislocation strengthening ($\Delta\sigma_{dis}$). $\sigma_0$ represents the baseline strength of pure aluminium, and $\sigma_{0.2}$ represents experimental values of the yield strength.

acquired at the same temperature. The Cu concentration for both conditions in the residual liquid is much higher than that in the solid network structure, and the peaks of the Cu concentrations are almost identical. However, the average width of the liquid channels for the nearest neighbor dendrites significantly increases with increasing dendrites size. This implies that more solute accumulates in the liquid channels, which suggests a higher segregation degree at a given temperature for larger dendrites size. Notably, strong segregation behavior in liquid channels can impair extensive coalescence of the dendrites and thus make the dendrite susceptible to solidification cracking[36].

To realize the quantitative description of SCS using Kou's index, $T$-$(f_s)^{1/2}$ curves are obtained from PF simulated data (Fig. 5e). Figure 5e compares the $T$-$(f_s)^{1/2}$ curves of the melting zone fabricated with the ER2319 and the ZCASW fillers. At the terminal solidification stage with high $f_s$, the $T$-$(f_s)^{1/2}$ curve corresponding to the ER2319 filler is steeper than that corresponding to the ZCASW filler. The SCS indexes proposed by Kou[35], i.e., the maximum $|dT/d(f_s)^{1/2}|$ near $(f_s)^{1/2} = 1$, are demonstrated in Fig. 5e. Note that the steeper slope of the $T$-$(f_s)^{1/2}$ curve near $(f_s)^{1/2} = 1$ indicates a higher cracking susceptibility[35]. Moreover, we plot the curves of $|dT/d(f_s^{1/2})|$ vs $T$ at the final solidification stage in the inset of Fig. 5e. It is evident that the cracking susceptibility index corresponding to the ER2319 material is higher than that corresponding to the ZCASW filler. Therefore, the melting zone fabricated

with the ER2319 filler has higher SCS than that fabricated with the ZCASW filler.

## Mechanical properties of welded joints

To reveal the influence of the ZCASW filler on the mechanical properties of the welded joint, we conduct the microhardness and tensile tests. Figure 6a compares the cross-sectional microhardness of the welds fabricated with the ER2319, the ER4043 and the ZCASW fillers. The microhardness of the weld fabricated with the ZCASW filler is obviously enhanced compared with those of the conventional fillers. This can be attributed to an altered chemical composition and commonly known strengthening mechanisms, such as the Hall-Petch effect[37].

Figure 6b shows that the average tensile strength of the welds fabricated with the ER2319 and ER4043 fillers is 318 MPa and 290 MPa, which is up to 72% and 66% of that of the parent materials, respectively. While the average tensile strength of the weld fabricated with the ZCASW filler reaches 349 MPa, which is considerably higher than those fabricated with the previous two fillers, i.e., ~79% of that of the parent material ($\sigma_b$ = 442 MPa). A summary of the ultimate tensile strengths of the ZCASW welds compared to the welds fabricated with ER2319, the ER4043 and other fillers can be found in Fig. 6c. The weld fabricated with the ZCASW exhibits the best strength, surpassing the majority of values reported in the literature[8]. Furthermore, the tensile strength of

the weld fabricated with the ZCASW filler without post-heat treatment is even comparable to that fabricated with the FSW ($\sigma_b$ = 360 MPa)[38]. To understand the origin of the increase in weld strength, we perform a quantitative analysis of the weld fabricated with the ZCASW filler according to a widely recognized strength mechanism, including grain size strengthening ($\Delta\sigma_{gb}$), solid solution strengthening ($\Delta\sigma_{ss}$), dislocation strengthening ($\Delta\sigma_{dis}$), and precipitation strengthening ($\Delta\sigma_{ppt}$). Table 1 shows the characteristics (radius $r$, and volume fraction $f$), grain sizes ($d$), solute concentrations ($c_{Cu}$ and $c_{Mg}$), and dislocation densities ($\rho$) of the melting zone. According to these characteristic values (see Methods), the strengthening contributions mentioned above can be calculated (see Methods). Considering that these strengthening mechanisms are effectively present at room temperature, it is reasonable to assume that each strength contribution could be linearly superimposed[39,40]. The relative contributions are displayed in Table 2. In addition, the intrinsic yield strength of pure aluminium ($\sigma_0$ = 10 MPa[41]) must be considered. Consequently, the overall increment for yield strength can be summarized as[42]:

$$\sigma_y = \sigma_0 + \Delta\sigma_{ss} + \Delta\sigma_{ppt} + \Delta\sigma_{gb} + \Delta\sigma_{dis} \qquad (1)$$

Figure 6d displays the linear superposition of the overall yield strength increments after welding, and the theoretical calculations are consistent with the experimental results. The grain size strengthening makes the most significant contribution to the melting zone at 47.2%, while precipitation strengthening is relatively insignificant. Solid solution strengthening also prevails in the melting zone, accounting for a contribution of 25.4%, and the remainder is dislocation strengthening, with a 21.2% contribution to the melting zone.

## Methods

### Filler material fabrication

Considering the effect of Zr on the AA2024 welds, six ER2319 wires with a diameter of 0.6 mm, and one R60702 wire with a diameter of 0.6 mm were twisted into 1.8 mm diameter ZCASW filler. The R60702 wire was used as the central wire, and six ER2319 wires were twisted and wound around the central wire. The ER2319 wires were the key component of the ZCASW filler due to their similar chemical components to the parent material. The main chemical component of the R60702 wire was Zr, which was prone to react with $\alpha$-Al to form $Al_3Zr$. Experimental studies[32] had shown that $Al_3Zr$ phases could act as heterogeneous nuclei and greatly refine the grains of the $\alpha$-Al. The nominal chemical compositions of the substrate and welding wire are listed in Table 3, and supplementary Table 3 lists the composition of the weld fabricated with the ZCASW filler, and the test was conduct using Inductive coupled plasma optical emission spectrometer (ICP-OES).

### Welding procedure and sample preparation

A hybrid oscillating laser-arc welding method was adopted to fuse 2024 Al alloy sheets. The ER2319 and ER4043 fillers with a 1.2 mm diameter (chemical composition can be seen in Table 3) and the ZCASW filler with a 1.8 mm diameter were applied in the experiments. Table 4 showed the optimized welding parameters. Figure 1a demonstrates the schematic diagram of welding process.

Afterwards, all samples were machined via wire electro discharge machining (EDM). Tensile specimens were cut to a thickness of 3 mm using EDM and the surfaces were polished with 240, 400, 800 grit sandpapers in preparation for mechanical testing. Microstructure samples were also cut with EDM and mounted in epoxy resin for polishing. Grinding was performed with 240, 400, 800 and 1200 grit sandpapers. Final polishing of the samples was accomplished with 1 μm diamond and 50 nm $Al_2O_3$ polishing compounds. Some polished samples were etched with Keller's Etchant for 15 s for SEM analysis. Additional samples were used for EBSD analysis. For TEM characterization, a sample of approximately 30 nm thickness was cut from the melting zone using a FIB microscope.

### X-ray micro computed tomography technology and materials characterization

To characterize the spatial distribution of the solidification crack defects, X-ray micro computed tomography technology was performed on a Zeiss Xradia 620. 2 mm × 2 mm × 5 mm microvolume samples were cut for X-ray microtomography. To identify typical minimum detectable defects during scanning, and 3 μm × 3 μm × 3 μm voxels were selected[43]. The visualization was conducted with an Avizo software.

For microstructural analysis, the polished samples were observed using an optical microscope and SEM. As for EBSD analysis, the samples after grinding were polished with 10% perchloric acid and alcohol solution, and then observed using a GeminiSEM300. The samples for EBSD analysis were further analyzed to determine the elemental distributions across the dendrites and interdendritic regions using an EPMA-8050G. For TEM analysis, both the composition of the precipitates within the dendrites and the interdendritic phases were examined by a Spectra 300 for the 30-nm thick samples.

### Mechanical testing

The microhardness tests were conducted at 2 mm from the upper surface of the welds, with machine settings of 100 g force and 15 s dwell time.

**Table 1 | Characteristic values of the microstructure for strength calculation in the melting zone (MZ)**

| Materials | Grain size | Solute concentrations | | Dislocation densities | Precipitation size and volume fraction | |
|---|---|---|---|---|---|---|
| | $d$ (μm) | $c_{Mg}$ (wt%) | $c_{Cu}$ (wt%) | $\rho_{GND}$ (m$^{-2}$) | $r$ (nm) | $f$ (%) |
| MZ | 4 | 2.12 | 2.02 | $1.5 \times 10^{14}$ | 617 | 2.44 |

**Table 2 | Estimates of the strengthening contributions in the melting zone**

| Materials | $\sigma_0$ (MPa) | $\Delta\sigma_{gb}$ (MPa) | $\Delta\sigma_{ppt}$ (MPa) | $\Delta\sigma_{ss}$(MPa) | $\Delta\sigma_{dis}$ (MPa) | $\Delta\sigma_y^{Cal}$ (MPa) |
|---|---|---|---|---|---|---|
| MZ | 10 | 125 | 6 | 67.3 | 56.3 | 265 |

**Table 3 | Chemical composition of substrate and welding wires/wt%**

| Composition | Si | Cu | Mg | Mn | Fe | Ti | Al | Zn | Zr |
|---|---|---|---|---|---|---|---|---|---|
| AA2024 | 0.5 | 3.9–4.9 | 1.2–1.8 | 0.3–0.9 | 0.5 | 0.15 | Bal | 0.25 | --- |
| ER2319 | 0.2 | 5.8–6.8 | 0.02 | 0.2–0.4 | 0.3 | 0.1–0.2 | Bal | 0.1 | 0.1–0.25 |
| ER4043 | 4.5–6.0 | 0.3 | 0.05 | 0.05 | 0.8 | 0.2 | Bal | 0.1 | --- |
| | Zr+Hf | Hf | Fe+Cr | Sn | H | | N | | O |
| R60702 | ≥99.2 | ≤4.5 | ≤0.2 | --- | ≤0.005 | | ≤0.025 | | ≤0.16 |

**Table 4 | Welding parameters**

| Parameter | Value |
|---|---|
| Beam wave length (nm) | 1064 |
| Focus spot diameter (mm) | 0.1 |
| Focal length of lens (mm) | 200 |
| Defocus distance (mm) | 0 |
| Laser power (kW) | 4.8 |
| Welding speed (mm s$^{-1}$) | 20 |
| Oscillating diameter (mm) | 1.5 |
| Oscillating frequency (Hz) | 250 |
| Electric current (A) | 30 |
| Voltage (V) | 14.5 |
| Argon flow rate (L min$^{-1}$) | 20 |

The monotonic tensile tests using an INSTRON-8801were conducted on three kinds of welds fabricated with the ER2319, the ER4043 and the ZCASW fillers, with a minimum width of 12 mm, a gage length of 40 mm and a thickness of 3 mm (Supplementary Fig. 7). The nominal strain rate was 2.0 mm/min. The loading direction was perpendicular to the welding direction.

**Phase field (PF) modeling**

The phase-field (PF) model developed by Takaki and Ohno[44,45] was adopted in this study. An important advantage of this model was that it supported solute diffusion in both the solid and liquid phases. Notably, the side branches were expressed by introducing a fluctuating current **J** of the Gaussian random number with the variance of $2D_l F_u^o q(\phi)(1+(1-k)u)/(\Delta t \Delta x^d)$, where $F_u^o$ represents the constant noise magnitude depending on the interfacial thickness, $\Delta t$ is the time step and $\Delta x$ is the lattice size.

An Al-4.5 wt% Cu alloy was chosen, and the physical properties was listed in supplementary Table 1[45–47]. The governing equations of the PF model were discretized by the normal finite-differential method. The time evolutions were discretized by the first-order Euler scheme. The time-step size was selected according to $\Delta t \le (\Delta x)^2/5.0D_l$. A uniform mesh spacing of $\Delta x = 0.8W_0 = 0.025$ μm was used, where $W_0$ refers to the interfacial thickness. The computational domain size was set as 75 μm × 75.4 μm (3000 × 3016 cells), which satisfied the actual requirements. For the cooling rate in this study, we used 2400 K·s$^{-1}$, which was obtained from the actual measurements (Supplementary Fig. 1). The simulations were accelerated by multiple CPU parallel computations using the message-passing interface library.

**Estimating the strengthening contributions**

According to the classical Hall-Petch equation, the strengthening contribution due to the grain boundaries could be given by the following equation[37]:

$$\Delta \sigma_{gb} = k \cdot d^{-1/2} \quad (2)$$

where $k$ is the strengthening coefficient specific to each material and $k = 0.12$ for aluminium alloy, $d$ is the grain size derived from the EBSD analysis (Fig. 3h, i).

According to the well-known Fleischer equation, the strengthening contribution arising from solute strengthening could be estimated using the following equation[48]:

$$\Delta \sigma_{ss} = \sum_i \Delta \sigma_i c_i \quad (3)$$

where $\Delta \sigma_i$ represents the theoretical strengthening efficiency for each element ($\Delta \sigma_{Mg} = 18.6$ MPa/wt.% and $\Delta \sigma_{Cu} = 13.8$ MPa/wt.%[48]). $c_i$ is the

concentration of the solute element (in wt.%), which were measured by SEM-EDS in the study (supplementary Table 2).

The strengthening contribution arising from dislocation strengthening was estimated using the Bailey-Hirsch equation[49]:

$$\Delta \sigma_{dis} = M \alpha G b \rho_{GND}^{1/2} \quad (4)$$

where $\alpha$ represents a constant, which is set as 0.2 for the face-centered cubic (fcc) alloy. $\rho_{GND}$ is the GND density, which was determined by EBSD analysis (supplementary Fig. 4).

The strength increments arising from the Orowan bypassing mechanism could be assessed as follows[21]:

$$\Delta \sigma_{pspt} = \frac{0.4MGb \ln \frac{r\pi}{2b}}{\pi \sqrt{1-\upsilon}} \frac{1}{L_p} \quad (5)$$

where $M = 3.06$ represents the Taylor factor of the fcc alloy system, $G = 27$ GPa is the shear modulus of aluminium alloy, $\upsilon = 0.33$ is Poisson's ratio of the Al alloy, $b = 0.286$ nm is the Burgers vector value of the Al alloy, $r$ is the mean precipitate radius. $L_p = r(\sqrt{\frac{2\pi}{3f}} - \frac{\pi}{2})$ represents the inter-precipitate distance, in which $f$ is the precipitate volume fraction. In our case, $r$ and $f$ were extracted from the SEM images using the image processing software (Supplementary Fig. 6).

## Data availability

The data that support the findings of this study are available from the corresponding author upon request. The source data for Figs. 4c, 5d, e, 6a, b, supplementary Fig. 2 and supplementary Fig. 3 used in this study are available in Zenodo with the identifier [https://doi.org/10.5281/zenodo.10562571][50].

## Code availability

The code that support the findings of this study are available from the corresponding author upon request.

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

## Acknowledgements

This research has been supported by the National Natural Science Foundation of China under Grant No.U22A20196 (P.J.), 52305467 (S.N.G.), 52075201 (P.J.), 52188102 (P.J.), GuangDong Basic and Applied Basic Research Foundation 2023A1515010081 (S.N.G.), and Fundamental Research Funds for the Central Universities under the Grant No. YCJJ20230360 (L.Y.R.).

## Author contributions

J.J., S.N.G and C.H. conceived the idea and designed the experiments. J.J. and S.N.G. fabricated the ZCASW filler material. J.J. and Y.T.L conducted the welding experiments. J.J., X.Q.W and conducted in X-ray micro CT experiments and analyzed the data. J.J. characterized microstructures. C.H., L.Y.R. and J.J. performed the phase field modeling and analyzed solidification cracking susceptibility. J.J. and L.Y. performed mechanical tests. J.J. and S.N.G. wrote the manuscript. P.J., L.S.S. and X.Y.S. supervised the whole work.

## Competing interests

The authors declare no competing interests.
