## [Peer Review File · Nature Communications]

High-strength and crack-free welding of 2024 aluminium alloy via novel Zr-core-Al-shell wireREVIEWER COMMENTS

Reviewer #1 (Remarks to the Author):

The article reports on the avoidance of cracking in high strength aluminium alloys through an innovative combination of aluminum and zirconium wires. As such, the idea is strong, the characterization detailed, but the science is not new. Therefore, I cannot recommend publication in a high-impact journal such as Nature Communications. Please see below detailed comments:

Abstract:

Line 11: The authors state that the Zr alloy was carefully selected. No reference is made to this selection process throughout the whole manuscript. Simply stating that Zr shows a grain refinement effect is insufficient and does not correspond to a selection process.

Line 16ff: The reviewer does not agree that this method will apply to other materials. There is body of literature that provides evidence for the grain refinement effect of Zr in aluminium, but this is not equivalently the case in other alloys. Thus, the method is less disruptive than suggested by the authors.

Introduction:

Line 44: This is not true; AA2024 is not a eutectic alloy.

Results:

Line 137: Based on the literature, it is actually no surprise that the approach chosen works.

Line 197: Please give the definition of Q. The reviewer believes that it is inconvenient for the reader to have to look it up elsewhere.

Methods:

Line 434: Can the authors give the chemical composition of the deposit? As the wires are of identical diameter, it appears that there is a lot of Zr in the deposit. This is an important information.

General comments: The authors state that AA2024 is not weldable, but recent research, which has not been cited, suggests the opposite, provided that advanced fusion welding techniques are applied.

English is of insufficient quality and there are several typing errors.

Further, it appears important that Zr is an expensive material. This aspect should be addressed somewhere, as the commercial viability of the approach is questionable.

Reviewer #2 (Remarks to the Author):

Achieving reliable and crack-free joints of AA2024 aluminum alloys by means of fusion welding is very challenging, as these alloy is susceptible to hot cracking. In order to overcome this issue, the present manuscript proposes the use of zirconium-core-aluminum-shell wires ('ZCASW') in hybrid laser-arc welding of AA2024. Although the presented topic is basically of high practical relevance, the current version of the manuscript is not suitable for being published in Nature Communications, which is due to the following reasons:

(1) English needs major revision.

Besides grammatical errors (e.g., incorrect use of was/were) and typing errors (e.g., 'phase fi[e]ld', line 493), the manuscript contains numerous misformulations, e.g.

- lines 32-33 "Welding or joining, dubbed 'industrial tailoring', is necessarily ubiquitous..."

- lines 126-127: "...characterized as a 'moderate' grain refiner identified with an ideal lattice matching that of Al to provide a low energy..."

- line 149: "Solidified microstructures have long been known to have a substantial influence on cracking during solidification", Do the authors would like to state that "The shape of the solid primary dendrites/crystals is known to influence..."

- line 178: "...the concentration of strengthening alloy AA2024 decreased", Do the authors would like to state that "The molten filler alloy locally dilutes the molten AA2024 base alloy."?

- lines 181-182: "The use of ZCASW filler material, an alternative method that has substantial potential for thoroughly altering the AA2024 solidification mechanism was employed successfully"; Do the authors would like to state that "The use of the ZCASW filler material successfully altered the solidification mechanism of the AA2024 alloy..."?
- line 246 "...melting zones welded with...", line 257 "...the melting zone with..."
- many more...

(2) The wording is not exact (e.g., When describing the microstructure, the terms 'grains' and 'grain boundaries' were frequently used instead of the terms 'dendrites' and 'interdendritic regions', respectively).

(3) The presented ZCASW was only used in hybrid arc-laser welding of AA2024. Therefore, the statement "This rather simple modification of the filler material of a fusion weld could be generally applied to a wide range of materials that are susceptible to hot cracking, such as nonwelded ['nonweldable!'] nickel superalloys..." is just speculation, as it has not been proven. The authors should avoid purely speculative statements (e.g., lines 17-24 and lines 93-99)!

(4) The first paragraph (lines 26-34) is intended to highlight the importance of lightweight design. However, it seems that the authors only cited secondary literature that is not related to the present topic instead of citing the relevant primary literature (e.g., original review papers) on weight reduction, fuel efficiency and lightweight materials.

(5) Hybrid laser-arc welding is not novel. Therefore, the introduction should also credit previous works on beam welding (electron and laser) and on hybrid welding of AA-2024.

(6) The introduction of the manuscript should contain fundamentals of grain refinement and epitaxial growth of aluminum alloys, as Figure 1 (c) shows a schematic illustration of the lattice match, but without any sufficient explanation. The reference(s) of the schematic illustration and of the crystallographic data are also missing.

(7) The description of hot cracking as given in lines 50-52 is not fully correct. The authors rather describe the formation of shrinkage cavities that form, if melt feeding is prevented during solidification. In simple words, hot cracking occurs during solidification of alloys, if the stress induced by solidification shrinkage locally exceeds the temperature-dependent tensile strength of the already solidified microstructure. As the strength of the solid interdendritic region is less than the strength of the primary dendrites, cracks predominantly form between the dendrites. However, if the feeding capability is sufficient, even these cracks may be fed with highly segregated melt that subsequently solidifies. Therefore, hot cracks do not necessarily appear as "empty" cavities in the final microstructure.

(8) The authors claim that the ZCASW weld did not contain any cracks. For confirmation they present a crack-free 3D reconstruction captured using computed tomography (CT). However, this confirmation is lacking for several reasons:

- The dimensions (L x B x H) of the cubic CT 'microvolume' as illustrated in Figure 1 are missing. If the dimensions of the CT volume are small in comparison to the dimensions of the weld, existing (macro)cracks may possibly be overlooked.

- The exact position of the CT volume inside of the weld seam should be provided. Did the authors extract the volume from the root, the top or the side section of the weld, or did they even extract it from the heat affected zone (HAZ)? Stating that "...a microvolume was excised from a region of [the] weld..." is not sufficient.

(9) It is supposed to separate Figure 1 (a, b and c) that illustrates the filler wire and the process from Figure 1 (d and e) that shows the weld seam. Moreover, clear macroscopic images of the weld cross-section and of the weld surface are lacking and must be added to the manuscript! The authors did not

present any high-resolution macroscopic image of the weld cross-section. Etching of the polished cross-section would be very beneficial to visualize the different microstructures of the weld, the HAZ and the base metal. Moreover, the cross-sections would enable to identify potential macrodefects, such as cracks or pores.

(10) The authors should be aware that Figure 2 (a, d, g) does not show any 'grains'. The bright areas obviously represent the primary dendritic phase (aluminum dendrites), the dark areas represent the interdendritic phase. The EDX element distribution maps in Figure 2 (k) and in Supplementary Figure 5 do not confirm that Zr, Cu and Mg are present in the weld, because only black squares are shown!

(11) Figure 3 (a, c) is unnecessary, as it does not provide any additional information to Figure 3 (b, d). Figure 3 (e, f) should be larger to provide valuable information. The authors claimed that an eutectic formed in the channels between the dendrites. Is it really an eutectic (i.e., it formed by coupled solidification of two phases) or is it just the remaining melt that was highly enriched in Cu, Mg and other elements and that finally solidified as low-melting phase? Are the authors sure about the crystallographic stoichiometry (S-Al₂CuMg)? Is there any evidence in the corresponding ternary phase diagrams?

(12) The authors should state where the values of the microstructural features that are summarized in Table 1 were taken from. Supplementary Table 1 should be part of the manuscript.

(13) Supplementary Figure 7 can be removed, as it does not provide any valuable information. The results of the presented spot measurements (high aluminum peak, low Cu and Mg peaks) are trivial, as all spots were located at the primary aluminum dendrites.

Reviewer #3 (Remarks to the Author):

This paper presents a new welding wire consisting of stranded aluminium wires around a Zr wire at the core to weld 'unweldable' 2024 Al alloy. This is a subject particularly interesting and relevant to the academic and industrial community. However, I am raising a number of concerns that need to be addressed before this work can be accepted for publication.

First and foremost, the new filler wire is proposed to avoid the formation of cracks in the weld but the authors provide no evidence of the cracks present using traditional filler material.

Fig. 1e shows a reconstructed 3D tomography image of the weld. The authors do comment on the absence of crack but it does seem that a number of voids are present. Can the authors comment on the importance of these voids? Are these comparable to what would be obtained in a traditional fusion weld? Would this distribution of voids be acceptable? It would have been interesting to perform a similar tomography experiment with traditional fillers for comparison.

Line 166 – The sentence does not really make sense, maybe 'if' needs to be removed

The authors mention the formation of cracks when using alternative fillers, particularly the ER2319 filler. However, no example of such cracks are shown. It would be beneficial to add such images of cracks to validate these claims.

Line 247 and 256. The higher magnification images are Fig 3b and Fig 3d (and not 3a and 3c)

The grain sizes from the EBSD in Fig 2 do not seem to match the ones from the microscopy presented in Fig 3. In Fig 3b and 3d, the grain size appears similar for the ER2319 and ZCASW.

It seems difficult to conclude on the morphology of the eutectic phases between fig 3b and 3d. It

seems that a large amount of the eutectic phases has been dislodged during the polishing step leaving a high number of voids at the GBs. So it seems difficult to conclude that 'the occurrence and size of the eutectic features were drastically reduced' from these images only.

Again, it is difficult to conclude anything from the CT provided in 3e as there is no discussion on the size and volume fraction of the eutectic features and crucially no comparison with the weld obtained with the alternative filler material.

The solidification behaviour is assessed via simulating the evolution of the liquid channel morphology between the two different fillers. A cracking susceptibility index is calculated as the slope of the T vs $f_s^{0.5}$ plot. The CSI is said to be higher for the ER2319 vs ZCASW wire. However, the curves are so close to each other that the difference between the two seems almost negligible. How relevant is the observed difference in CSI. Also, could you plot the derivative of T vs $f_s^{0.5}$ which would make it easier to conclude on whether there is a clear difference. At the moment it seems difficult to conclude from the information provided.

The mechanical properties of the different joints are then analysed using a combination of hardness and tensile tests.

The way the tensile strength is reported is confusing, do you quote the highest obtained strength and then in brackets the average and standard deviation? It would be more standard to only report the average value. How many tensile tests were conducted per condition?

When assessing the different contributions to strengthening, how were the precipitates analysed and what precipitates are we talking about? Is that the coarse Al_3Zr phases?

**REVIEWER COMMENTS**

Reviewer #1 (Remarks to the Author):

The article reports on the avoidance of cracking in high strength aluminium alloys
through an innovative combination of aluminum and zirconium wires. As such, the idea
is strong, the characterization detailed, but the science is not new. Therefore, I cannot
recommend publication in a high-impact journal such as Nature Communications.
Please see below detailed comments:

Abstract:

Line 11: The authors state that the Zr alloy was carefully selected. No reference is made
to this selection process throughout the whole manuscript. Simply stating that Zr shows
a grain refinement effect is insufficient and does not correspond to a selection process.

Line 16ff: The reviewer does not agree that this method will apply to other materials.
There is body of literature that provides evidence for the grain refinement effect of Zr
in aluminium, but this is not equivalently the case in other alloys. Thus, the method is
less disruptive than suggested by the authors.

Introduction:

Line 44: This is not true; AA2024 is not a eutectic alloy.

Results:

Line 137: Based on the literature, it is actually no surprise that the approach chosen
works.

Line 197: Please give the definition of Q. The reviewer believes that it is inconvenient
for the reader to have to look it up elsewhere.

Methods:

Line 434: Can the authors give the chemical composition of the deposit? As the wires
are of identical diameter, it appears that there is a lot of Zr in the deposit. This is an
important information.

General comments: The authors state that AA2024 is not weldable, but recent research,
which has not been cited, suggests the opposite, provided that advanced fusion welding
techniques are applied.

English is of insufficient quality and there are several typing errors.

Further, it appears important that Zr is an expensive material. This aspect should be
addressed somewhere, as the commercial viability of the approach is questionable.

Reviewer #2 (Remarks to the Author):

Achieving reliable and crack-free joints of AA2024 aluminum alloys by means of
fusion welding is very challenging, as these alloy is susceptible to hot cracking. In order
to overcome this issue, the present manuscript proposes the use of zirconium-core-
aluminum-shell wires ('ZCASW') in hybrid laser-arc welding of AA2024. Although

the presented topic is basically of high practical relevance, the current version of the
manuscript is not suitable for being published in Nature Communications, which is due
to the following reasons:

(1) English needs major revision.

Besides grammatical errors (e.g., incorrect use of was/were) und typing errors (e.g.,
'phase fi[e]ld', line 493), the manuscript contains numerous misformulations, e.g.

- lines 32-33 "Welding or joining, dubbed 'industrial tailoring', is necessarily
ubiquitous..."

- lines 126-127: "...characterized as a 'moderate' grain refiner identified with an ideal
lattice matching that of to Al to provide a low energy..."

- line 149: "Solidified microstructures have long been known to have a substantial
influence on cracking during solidification", Do the authors would like to state that
"The shape of the solid primary dendrites/crystals is known to influence..."

- line 178: "...the concentration of strengthening alloy AA2024 decreased", Do the
authors would like to state that "The molten filler alloy locally dilutes the molten
AA2024 base alloy."?

- lines 181-182: "The use of ZCASW filler material, an alternative method that has
substantial potential for thoroughly altering the AA2024 solidification mechanism was
employed successfully", Do the authors would like to state that "The use of the ZCASW
filler material succesfully altered the solidification mechanism of the AA2024 alloy..."?

- line 246 "...melting zones welded with...", line 257 "...the melting zone with..."

- many more...

(2) The wording is not exact (e.g., When describing the microstructure, the terms
'grains' and 'grain boundaries' were frequently used instead of the terms 'dendrites'
and 'interdendritic regions', respectively).

(3) The presented ZCASW was only used in hybrid arc-laser welding of AA2024.
Therefore, the statement "This rather simple modification of the filler material of a
fusion weld could be generally applied to a wide range of materials that are susceptible
to hot cracking, such as nonwelded ['nonweldable!'] nickel superalloys..." is just
speculation, as it has not been proven. The authors should avoid purely speculative
statements (e.g., lines 17-24 and lines 93-99)!

(4) The first paragraph (lines 26-34) is intended to highlight the importance of
lightweight design. However, it seems that the authors only cited secondary literature
that is not related to the present topic instead of citing the relevant primary literature
(e.g., original review papers) on weight reduction, fuel efficiency and lightweight
materials.

(5) Hybrid laser-arc welding is not novel. Therefore, the introduction should also credit
previous works on beam welding (electron and laser) and on hybrid welding of AA-
2024.

(6) The introduction of the manuscript should contain fundamentals of grain refinement
and epitaxial growth of aluminum alloys, as Figure 1 (c) shows a schematic illustration
of the lattice match, but without any sufficient explanation. The reference(s) of the
schematic illustration and of the crystallographic data are also missing.

(7) The description of hot cracking as given in lines 50-52 is not fully correct. The
authors rather describe the formation of shrinkage cavities that form, if melt feeding is
prevented during solidification. In simple words, hot cracking occurs during
solidification of alloys, if the stress induced by solidification shrinkage locally exceeds
the temperature-dependent tensile strength of the already solidified microstructure. As
the strength of the solid interdendritic region is less than the strength of the primary
dendrites, cracks predominantly form between the dendrites. However, if the feeding
capability is sufficient, even these cracks may be fed with highly segregated melt that
subsequently solidifies. Therefore, hot cracks do not necessarily appear as “empty”
cavities in the final microstructure.

(8) The authors claim that the ZCASW weld did not contain any cracks. For
confirmation they present a crack-free 3D reconstruction captured using computed
tomography (CT). However, this confirmation is lacking for several reasons:

- The dimensions (L x B x H) of the cubic CT ‘microvolume’ as illustrated in Figure 1
are missing. If the dimensions of the CT volume are small in comparison to the
dimensions of the weld, existing (macro)cracks may possibly be overlooked.

- The exact position of the CT volume inside of the weld seam should be provided. Did
the authors extract the volume from the root, the top or the side section of the weld, or
did they even extract it from the heat affected zone (HAZ)? Stating that “...a
microvolume was excised from a region of [the] weld...” is not sufficient.

(9) It is supposed to separate Figure 1 (a, b and c) that illustrates the filler wire and the
process from Figure 1 (d and e) that shows the weld seam. Moreover, clear macroscopic
images of the weld cross-section and of the weld surface are lacking and must be added
to the manuscript! The authors did not present any high-resolution macroscopic image
of the weld cross-section. Etching of the polished cross-section would be very
beneficial to visualize the different microstructures of the weld, the HAZ and the base
metal. Moreover, the cross-sections would enable to identify potential macrodefects,
such as cracks or pores.

(10) The authors should be aware that Figure 2 (a, d, g) does not show any ‘grains’. The
bright areas obviously represent the primary dendritic phase (aluminum dendrites), the
dark areas represent the interdendritic phase. The EDX element distribution maps in
Figure 2 (k) and in Supplementary Figure 5 do not confirm that Zr, Cu and Mg are
present in the weld, because only black squares are shown!

(11) Figure 3 (a, c) is unnecessary, as it does not provide any additional information to

Figure 3 (b, d). Figure 3 (e, f) should be larger to provide valuable information. The
authors claimed that an eutectic formed in the channels between the dendrites. Is it
really an eutectic (i.e., it formed by coupled solidification of two phases) or is it just the
remaining melt that was highly enriched in Cu, Mg and other elements and that finally
solidified as low-melting phase? Are the authors sure about the crystallographic
stoichiometry (S-Al₂CuMg)? Is there any evidence in the corresponding ternary phase
diagrams?

(12) The authors should state where the values of the microstructural features that are
summarized in Table 1 were taken from. Supplementary Table 1 should be part of the
manuscript.

(13) Supplementary Figure 7 can be removed, as it does not provide any valuable
information. The results of the presented spot measurements (high aluminum peak, low
Cu and Mg peaks) are trivial, as all spots were located at the primary aluminum
dendrites.

Reviewer #3 (Remarks to the Author):

This paper presents a new welding wire consisting of stranded aluminium wires around
a Zr wire at the core to weld 'unweldable' 2024 Al alloy. This is a subject particularly
interesting and relevant to the academic and industrial community. However, I am
raising a number of concerns that need to be addressed before this work can be accepted
for publication.

First and foremost, the new filler wire is proposed to avoid the formation of cracks in
the weld but the authors provide no evidence of the cracks present using traditional
filler material.

Fig. 1e shows a reconstructed 3D tomography image of the weld. The authors do
comment on the absence of crack but it does seem that a number of voids are present.
Can the authors comment on the importance of these voids? Are these comparable to
what would be obtained in a traditional fusion weld? Would this distribution of voids
be acceptable? It would have been interesting to perform a similar tomography
experiment with traditional fillers for comparison.

Line 166 – The sentence does not really make sense, maybe 'if' needs to be removed

The authors mention the formation of cracks when using alternative fillers, particularly
the ER2319 filler. However, no example of such cracks are shown. It would be
beneficial to add such images of cracks to validate these claims.

Line 247 and 256. The higher magnification images are Fig 3b and Fig 3d (and not 3a

and 3c)

The grain sizes from the EBSD in Fig 2 do not seem to match the ones from the
microscopy presented in Fig 3. In Fig 3b and 3d, the grain size appears similar for the
ER2319 and ZCASW.

It seems difficult to conclude on the morphology of the eutectic phases between fig 3b
and 3d. It seems that a large amount of the eutectic phases has been dislodged during
the polishing step leaving a high number of voids at the GBs. So it seems difficult to
conclude that ‘ the occurrence and size of the eutectic features were drastically reduced’
from these images only.

Again, it is difficult to conclude anything from the CT provided in 3e as there is no
discussion on the size and volume fraction of the eutectic features and crucially no
comparison with the weld obtained with the alternative filler material.

The solidification behaviour is assessed via simulating the evolution of the liquid
channel morphology between the two different fillers. A cracking susceptibility index
is calculated as the slope of the T vs $f_s^{0.5}$ plot. The CSI is said to be higher for the
ER2319 vs ZCASW wire. However, the curves are so close to each other that the
difference between the two seems almost negligible. How relevant is the observed
difference in CSI. Also, could you plot the derivative of T vs $f_s^{0.5}$ which would make
it easier to conclude on whether there is a clear difference. At the moment it seems
difficult to conclude from the information provided.

The mechanical properties of the different joints are then analysed using a combination
of hardness and tensile tests.

The way the tensile strength is reported is confusing, do you quote the highest obtained
strength and then in brackets the average and standard deviation? It would be more
standard to only report the average value. How many tensile tests were conducted per
condition?

When assessing the different contributions to strengthening, how were the precipitates
analysed and what precipitates are we talking about? Is that the coarse Al_3Zr phases?

Detailed Response to Reviewers' Comments

“High-strength and crack-free welding of unweldable aluminium alloys via novel Zr-core-Al-shell wires”

We very much appreciate the reviewer's comments and suggestions (listed in black type below), which have significantly helped us improve our paper. We have made substantial revisions to the original manuscript with all these suggestions incorporated. The detailed responses and the revisions made are given in blue and green type below. The resulting changes are highlighted in orange in the revised manuscript.

Best regards, and thanks.

Response to Referee #1:

General Comments:

The article reports on the avoidance of cracking in high strength aluminium alloys through an innovative combination of aluminum and zirconium wires. As such, the idea is strong, the characterization detailed, but the science is not new. Therefore, I cannot recommend publication in a high-impact journal such as Nature Communications. Please see below detailed comments:

Response: Thanks for your constructive comments. We are very sorry that we do not make the science of the article clear. Here, we conclude the science of the article as follows.

(1) A novel zirconium-core-aluminium-shell wire is invented and the oscillating laser-arc hybrid welding technique is adopted to synergistically control solidification during welding. Reliable and crack-free welding of 2024 aluminium alloy is achieved ultimately.

(2) The underlying mechanism why the zirconium-core-aluminium-shell wire could inhibit solidification cracking significantly during welding of AA2024 is revealed

from the perspective of dendritic structure and interdendritic phase.

(3) An approach combining the phase field simulation and Kou's criterion is
developed to reveal how the microstructure morphology and solute segregation affect
solidification cracking, and the solidification cracking susceptibility during welding is
quantitatively predicted.

(4) The influence mechanism of ZCASW filler on the mechanical properties of
welded joint is revealed, and the strength contribution is quantitatively calculated to
demonstrate why welded joint fabricated with ZCASW filler has higher tensile strength.

**Specific Comments:**

1. Line 11: The authors state that the Zr alloy was carefully selected. No reference is
made to this selection process throughout the whole manuscript. Simply stating that
Zr shows a grain refinement effect is insufficient and does not correspond to a
selection process.

**Response:** Thank you very much for your comments. We have added a discussion of
the selection process to the introduction of the revised manuscript, as follows.

"Introducing nucleation particles to produce identical ultrafine equiaxed structure has
been an effective method. It can enlarge the equiaxed region of the thermal gradient-
growth velocity curve¹, which could easily assist to generate equiaxed microstructure².
Furthermore, the emergence of nucleation particles could increase the undercooling at
the solid/liquid interface and decrease the critical nuclear radius³, thereby effectively
facilitating the grain refinement during solidification. To obtain ultrafine equiaxed
microstructures, the nucleation particles need to have similar lattice parameters to α -
Al³⁻⁵. Thus, common elements for inoculation treatments, such as Zr, Ti, and Sc, have

-
1. Todaro, C. J. et al. Grain structure control during metal 3D printing by high-intensity ultrasound. *Nat. Commun.* **11**, 1-9 (2020).
 2. Nie, X. J. et al. Effect of Zr content on formability, microstructure and mechanical properties of selective laser melted Zr modified Al-4.24Cu-1.97Mg-0.56Mn alloys. *J. Alloy. Compd.* **764**, 977-986 (2018).
 3. Atamanenko, T. V. et al. On the mechanism of grain refinement in Al-Zr-Ti alloys. *J. Alloy. Compd.* **509**, 57-60 (2011).

been chosen to form Al_3X ($X = \text{Zr, Ti, or Sc}$) owing to their small lattice mismatch with
$\alpha\text{-Al}$. The lattice parameter of Al_3Zr , Al_3Sc and Al_3Ti is 4.08 Å, 4.103 Å and 3.967 Å,
which is similar to that of Al of 4.049 Å⁴. Compared to Al_3Sc or Al_3Ti phase, the lower
misfit value (0.765%) between the Al_3Zr phase and $\alpha\text{-Al}$ phase can decrease the
nucleation barrier for precipitation⁵. Moreover, the Al_3Zr phase can serve as excellent
heterogeneous nucleation sites of $\alpha\text{-Al}$ due to the close structural resemblances between
the two phases⁶. For example, two kinds of cube-on-cube orientation relationships (OR)
exist between $\text{L1}_2\text{-Al}_3\text{Zr}$ and the $\alpha\text{-Al}$: $\text{Al}(001)//\text{L1}_2\text{-Al}_3\text{Zr}(001)$, $\text{Al}[110]//\text{L1}_2\text{-}$
$\text{Al}_3\text{Zr}[110]$ and $\text{Al}(100)//\text{L1}_2\text{-Al}_3\text{Zr}(100)$, $\text{Al}[010]//\text{L1}_2\text{-Al}_3\text{Zr}[010]$ ⁷.”

2. Line 16ff: The reviewer does not agree that this method will apply to other materials.
There is body of literature that provides evidence for the grain refinement effect of
Zr in aluminium, but this is not equivalently the case in other alloys. Thus, the
method is less disruptive than suggested by the authors.

**Response:** We are very sorry for our inaccurate representation. Here, we simply want
to express that other materials sensitive to solidification cracking can be processed
using the filler with novel structure instead of specific zirconium (Zr)-core-aluminium
(Al)-shell-wire (ZCASW). For example, Haynes 230 exhibits obvious cracking during
laser additive manufacturing, Zhao et al⁸ introduced a continuous uniform
interdendritic liquid film in the terminal stage of solidification by increasing Zr content
and eliminated the hot cracking during the laser additive manufacturing. Therefore, we
could choose the combination of ERNiCrWMo-1 and R60702 welding wire to fabricate

-
4. Saha, S., Todorova, T. Z. & Zwanziger, J. W. Temperature dependent lattice misfit and coherency of Al_3X ($X = \text{Sc, Zr, Ti and Nb}$) particles in an Al matrix. *Acta Mater.* **89**, 109-115 (2015).
 5. Jiang, S. H. et al. Ultrastrong steel via minimal lattice misfit and high-density nanoprecipitation. *Nature.* **544**, 460-464 (2017).
 6. Prasad, K. S. et al. On the formation of faceted Al_3Zr (beta') precipitates in Al-Li-Cu-Mg-Zr alloys. *Acta Mater.* **47**, 2581-2592 (1999).
 7. Srinivasan, D. & Chattopadhyay, K. Non-equilibrium transformations involving $\text{L1}_2\text{-Al}_3\text{Zr}$ in ternary Al-X-Zr alloys. *Metall. Mater. Trans. A.* **36**, 311-320 (2005).
 8. Zhao, Y. N. et al. New alloy design approach to inhibiting hot cracking in laser additive manufactured nickel-based superalloys. *Acta Mater.* **247**, 118736 (2023).

the novel filler for Haynes 230 additive manufacturing. Taking the above into
considerations, we would make changes to the relevant parts.

“This ZCASW provides a foundation for broad industrial applications because it meets
the demands for efficiency in automated welding. This technology (the ZCASW
coupled with oscillating laser-arc hybrid welding) also has a great potential for metal-
based additive manufacturing of high-strength aluminium alloys, in which
solidification cracking is a common issue.”

3. Line 44: This is not true; AA2024 is not a eutectic alloy.

**Response:** This sentence has been corrected in the revised manuscript.

4. Line 137: Based on the literature, it is actually no surprise that the approach chosen
works.

**Response:** We have corrected this sentence in the revised manuscript, as follows.

“Fig. 2g exhibits no cracks as well, indicating that this novel filler material is
remarkably effective.”

5. Line 197: Please give the definition of Q. The reviewer believes that it is
inconvenient for the reader to have to look it up elsewhere.

**Response:** We have added the definition of Q to the revised manuscript, as follows.

“the growth-restricting factor (Q), can be expressed as follow⁹:

$$Q=m_lC_0(k-1)$$

Here, m_l represents the gradient of liquidus, C_0 represents the initial composition of
alloy, and k represents the partition coefficient.”

6. Line 434: Can the authors give the chemical composition of the deposit? As the
wires are of identical diameter, it appears that there is a lot of Zr in the deposit. This
is an import information.

**Response:** Thank you very much for your valuable comments. We have added the

9. Li, R. D. et al. Developing a high-strength Al-Mg-Si-Sc-Zr alloy for selective laser melting: Crack-inhibiting and multiple strengthening mechanisms. *Acta Mater.* **193**, 83-98 (2020).

chemical composition of deposit fabricated with the ZCASW filler material to the
revised manuscript, and the test is conducted using Inductive Coupled Plasma Optical
Emission Spectrometer (ICP-OES). The results are shown in Table 1, and the Zr content
in the deposit is 1.8066%.

Table 1 Chemical composition of deposit fabricated with the ZCASW filler/wt%

Composition	Al	Cu	Mg	Ti	Mn	Zr
Deposit	91.7117	4.1957	1.4899	0.0653	0.4934	1.8066

7. The authors state that AA2024 is not weldable, but recent research, which has not
been cited, suggests the opposite, provided that advanced fusion welding techniques
are applied.

**Response:** We are very sorry for our inaccurate statements. We just want to state that
the 2024 aluminum alloy is hard-to-weld. The argument that AA2024 has poor
weldability has been stated by leading welding companies such as Lincoln Electric and
ESAB. Please see the following website¹⁰⁻¹¹. Taking the above into considerations, we
will change the title of “High-strength and crack-free welding of unweldable aluminium
alloys via novel Zr-core-Al-shell wires” into “High-strength and crack-free welding of
2024 aluminium alloy via novel Zr-core-Al-shell wire”.

8. English is of insufficient quality and there are several typing errors.

**Response:** We have carefully modified the language of the article and corrected typing
errors. Meanwhile, we have invited professionals to polish the language of the article.
The revised parts are highlighted in orange in the revised manuscript.

9. Further, it appears important that Zr is an expensive material. This aspect should be
addressed somewhere, as the commercial viability of the approach is questionable.

**Response:** Thanks for your constructive comments. We agree with your opinion. As
you point out, Zr is a relatively expensive material, which increases the cost of the

10. https://esab.com/us/nam_en/esab-university/blogs/how-do-i-weld-2024-and-7075/

11. <https://www.lincolnelectric.com/en-us/support/welding-solutions/Pages/aluminum-faqs-detail.aspx#question8>

ZCASW filler material. We have estimated the cost of the filler material. The price of
producing 1 kg ZCASW filler material is about ¥ 704 CNY and the price of 1 kg
ER2319 filler material is about ¥ 330 CNY. In the future, we will optimize the structure
of our ZCASW filler material and reduce zirconium content, which is help to further
decrease the cost.

**Response to Referee #2:**

**General Comments:**

Achieving reliable and crack-free joints of AA2024 aluminum alloys by means of
fusion welding is very challenging, as these alloy is susceptible to hot cracking. In order
to overcome this issue, the present manuscript proposes the use of zirconium-core-
aluminum-shell wires ('ZCASW') in hybrid laser-arc welding of AA2024. Although
the presented topic is basically of high practical relevance, the current version of the
manuscript is not suitable for being published in Nature Communications, which is due
to the following reasons:

**Specific Comments:**

1. English needs major revision.

Besides grammatical errors (e.g., incorrect use of was/were) and typing errors (e.g.,
'phase fi[e]ld', line 493), the manuscript contains numerous misformulations, e.g.

**Response:** We have carefully modified the language of the article and corrected typing
errors. Meanwhile, we have invited professionals to polish the language of the article.
As for the numerous misformulations, we also have already made great changes in the
revised manuscript. The revised parts are highlighted in orange in the revised
manuscript. We have revised each of the misformulations that have been listed below.

1)- lines 32-33 "Welding or joining, dubbed 'industrial tailoring', is necessarily
ubiquitous..."

**Response:** This sentence has been corrected as follows.

"Welding is an important process for assembling lightweight materials."

2) - lines 126-127: "...characterized as a 'moderate' grain refiner identified with an
ideal lattice matching that of Al to provide a low energy..."

**Response:** We have corrected this sentence in the revised manuscript, as follows.

"This phase could significantly improve grain refinement efficiency and provide a low-
energy nucleation barrier for α -Al according to classical nucleation theory^{3,5}."

3) - line 149: "Solidified microstructures have long been known to have a
substantial influence on cracking during solidification", Do the authors would like
to state that "The shape of the solid primary dendrites/crystals is known to
influence..."

**Response:** Thanks for your comments. This sentence has been corrected, as shown
below.

"The shape of the solid primary dendrites/crystals is known to influence cracking
susceptibility during solidification."

4) - line 178: "...the concentration of strengthening alloy AA2024 decreased", Do
the authors would like to state that "The molten filler alloy locally dilutes the molten
AA2024 base alloy."?

**Response:** We have corrected this sentence in the revised manuscript, as follows.

"The molten filler alloy locally dilutes the molten AA2024 base alloy, which reduces
the concentration of strengthening alloy⁶."

5) - lines 181-182: "The use of ZCASW filler material, an alternative method that
has substantial potential for thoroughly altering the AA2024 solidification
mechanism was employed successfully", Do the authors would like to state that
"The use of the ZCASW filler material successfully altered the solidification
mechanism of the AA2024 alloy..."?

**Response:** This sentence has been corrected in the revised manuscript, as shown below.

"The use of the ZCASW filler material successfully altered the solidification
mechanism during welding of the AA2024 alloy."

6) - line 246 "...melting zones welded with...", line 257 "...the melting zone with..."

-
3. Atamanenko, T. V. et al. On the mechanism of grain refinement in Al-Zr-Ti alloys. *J. Alloy. Compd.* **509**, 57-60 (2011).
 5. Jiang, S. H. et al. Ultrastrong steel via minimal lattice misfit and high-density nanoprecipitation. *Nature.* **544**, 460-464 (2017).
 6. Sokoluk, M. et al. Nanoparticle-enabled phase control for arc welding of unweldable aluminum alloy 7075. *Nat. Commun.* **10**, 1-8 (2019).

- many more...

**Response:** These sentences have been corrected as follows.

“Fig. 3a presents the melting zone fabricated with the ER2319 filler material.”

“...the melting zone fabricated with the ER2319 and the ZCASW filler materials...”

2. The wording is not exact (e.g., When describing the microstructure, the terms
‘grains’ and ‘grain boundaries’ were frequently used instead of the terms ‘dendrites’
and ‘interdendritic regions’, respectively).

**Response:** We have corrected the statements about describing the microstructure, as
follows.

“We conducted microstructure analysis from the perspective of the dendrite
morphology and the interdendritic region morphology in the melting zones fabricated
with different fillers.”

“Here, we first discuss the effect of the dendrite morphology in detail. Fig. 3a presents
the melting zone fabricated with the ER2319 filler material.”

3. The presented ZCASW was only used in hybrid arc-laser welding of AA2024.
Therefore, the statement “This rather simple modification of the filler material of a
fusion weld could be generally applied to a wide range of materials that are
susceptible to hot cracking, such as nonwelded [‘nonweldable!’] nickel
superalloys....” is just speculation, as it has not been proven. The authors should
avoid purely speculative statements (e.g., lines 17-24 and lines 93-99)!

**Response:** Thank you very much for your insightful comments. We have corrected the
corresponding statements, as shown below.

“This ZCASW provides a foundation for broad industrial applications because it meets
the demands for efficiency in automated welding. This technology (the ZCASW
coupled with oscillating laser-arc hybrid welding) also has a great potential for metal-
based additive manufacturing of high-strength aluminium alloys, in which
solidification cracking is a common issue.”

“This new welding technology provides a foundation for broad industrial applications

because it could meet the demands for efficiency in automated welding.”

4. The first paragraph (lines 26-34) is intended to highlight the importance of
lightweight design. However, it seems that the authors only cited secondary
literature that is not related to the present topic instead of citing the relevant primary
literature (e.g., original review papers) on weight reduction, fuel efficiency and
lightweight materials.

**Response:** Thank you very much for your comments. We have cited the original review
papers on weight reduction, fuel efficiency and lightweight materials in the introduction,
as follows.

“Today, lightweight materials are an important component in promoting energy and
environmental sustainability^{12,13}. Every additional 100 kg decrease in vehicle weight
leads to a reduction in CO₂ emissions of 8.7 g per kilometre and fuel consumption of
0.4 litres per 100 kilometres¹⁴.”

5. Hybrid laser-arc welding is not novel. Therefore, the introduction should also credit
previous works on beam welding (electron and laser) and on hybrid welding of AA-
2024.

**Response:** We have added the corresponding contents about electron beam welding, arc
welding, laser beam welding and laser-arc hybrid welding of AA2024 to the
introduction in the revised manuscript, as shown below.

“While fusion welding is undoubtedly more flexible and efficient, numerous scholars
have conducted research on the welding of AA2024, such as arc welding¹⁵, laser

12. Zhang, W. & Xu, J. Advanced lightweight materials for Automobiles: A review. *Mater. Des.* **221**, 110994 (2022).

13. Lu, K. The Future of Metals. *Science*. **328**, 319-320 (2010).

14. Bandivadekar A.E.A. On the Road in 2035—Reducing Transportation’s Petroleum Consumption and GHG Emissions. Massachusetts Institute of Technology; Cambridge, MA, USA: 2008.

15. Soysal, T. & Kou, S. Effect of filler metals on solidification cracking susceptibility of Al alloys 2024 and 6061. *J. Mater. Process. Technol.* **266**, 421-428 (2019).

16. Sheikhi, M., Malek Ghaini, F. & Assadi, H. Prediction of solidification cracking in pulsed laser welding of 2024 aluminum alloy. *Acta Mater.* **82**, 491-502 (2015).

welding¹⁶, electron beam welding¹⁷ and hybrid welding¹⁸. And they highlighted that
one of the primary problems of fusion welding is solidification cracking, which
considerably hinders its widespread use.”

6. The introduction of the manuscript should contain fundamentals of grain refinement
and epitaxial growth of aluminum alloys, as Figure 1 (c) shows a schematic
illustration of the lattice match, but without any sufficient explanation. The
reference(s) of the schematic illustration and of the crystallographic data are also
missing.

**Response:** Thank you for your valuable comment. We have added the fundamentals of
grain refinement and epitaxial growth of aluminum alloys to the introduction of the
revised manuscript, and the references of the schematic illustration and the
crystallographic data have also been added as follows.

“Introducing nucleation particles to produce identical ultrafine equiaxed structure has
been an effective method. It can enlarge the equiaxed region of the thermal gradient-
growth velocity curve¹, which could easily assist to generate equiaxed microstructure².
Furthermore, the emergence of nucleation particles could increase the undercooling at
the solid/liquid interface and decrease the critical nuclear radius³, thereby effectively
facilitating the grain refinement during solidification. To obtain ultrafine equiaxed
microstructures, the nucleation particles need to have similar lattice parameters to α -

17. Hosseini, S. A. et al. Elimination of hot cracking in the electron beam welding of AA2024-T351 by controlling the welding speed and heat input. *J. Manuf. Process.* **46**, 147-158 (2019).

18. Yan, J. et al. Effect of welding wires on microstructure and mechanical properties of 2A12 aluminum alloy in CO₂ laser-MIG hybrid welding. *Appl. Surf. Sci.* **255**, 7307-7313 (2009).

1. Todaro, C. J. et al. Grain structure control during metal 3D printing by high-intensity ultrasound. *Nat. Commun.* **11**, 1-9 (2020).

2. Nie, X. J. et al. Effect of Zr content on formability, microstructure and mechanical properties of selective laser melted Zr modified Al-4.24Cu-1.97Mg-0.56Mn alloys. *J. Alloy. Compd.* **764**, 977-986 (2018).

3. Atamanenko, T. V. et al. On the mechanism of grain refinement in Al-Zr-Ti alloys. *J. Alloy. Compd.* **509**, 57-60 (2011).

4. Saha, S., Todorova, T. Z. & Zwanziger, J. W. Temperature dependent lattice misfit and coherency of Al₃X (X = Sc, Zr, Ti and Nb) particles in an Al matrix. *Acta Mater.* **89**, 109-115 (2015).

Al³⁻⁵. Thus, common elements for inoculation treatments, such as Zr, Ti, and Sc, have
been chosen to form Al₃X (X= Zr, Ti, or Sc) owing to their small lattice mismatch with
α-Al. The lattice parameter of Al₃Zr, Al₃Sc and Al₃Ti is 4.08 Å, 4.103 Å and 3.967 Å,
respectively, which is similar to that of Al of 4.049 Å⁴. Compared to Al₃Sc or Al₃Ti
phase, the lower misfit value (0.765%) between the Al₃Zr phase and α-Al phase can
decrease the nucleation barrier for precipitation⁵. Moreover, the Al₃Zr phase can serve
as excellent heterogeneous nucleation sites of α-Al due to the close structural
resemblances between the two phases⁶. For example, two kinds of cube-on-cube
orientation relationships (OR) exist between L1₂-Al₃Zr and the α-Al: Al(001)//L1₂-
Al₃Zr(001), Al[110]//L1₂-Al₃Zr[110] and Al(100)//L1₂-Al₃Zr(100), Al[010]//L1₂-
Al₃Zr[010]⁷.”
“Fig. 1c displays the crystallographic data of α-Al and L1₂-Al₃Zr and their lattice match
relationships^{7,19,20}.”

7. The description of hot cracking as given in lines 50-52 is not fully correct. The
authors rather describe the formation of shrinkage cavities that form, if melt feeding
is prevented during solidification. In simple words, hot cracking occurs during
solidification of alloys, if the stress induced by solidification shrinkage locally
exceeds the temperature-dependent tensile strength of the already solidified
microstructure. As the strength of the solid interdendritic region is less than the
strength of the primary dendrites, cracks predominantly form between the dendrites.
However, if the feeding capability is sufficient, even these cracks may be fed with
highly segregated melt that subsequently solidifies. Therefore, hot cracks do not

-
5. Jiang, S. H. et al. Ultrastrong steel via minimal lattice misfit and high-density nanoprecipitation. *Nature*. **544**, 460-464 (2017).
 6. Prasad, K. S. et al. On the formation of faceted Al₃Zr (beta') precipitates in Al-Li-Cu-Mg-Zr alloys. *Acta Mater.* **47**, 2581-2592 (1999).
 7. Srinivasan, D. & Chattopadhyay, K. Non-equilibrium transformations involving L1₂-Al₃Zr in ternary Al-X-Zr alloys. *Metall. Mater. Trans. A*. **36**, 311-320 (2005).
 19. Martin, J. H. et al. 3D printing of high-strength aluminium alloys. *Nature*. **549**, 365-369 (2017).
 20. Murty, B. S., Kori, S. A. & Chakraborty, M. Grain refinement of aluminium and its alloys by heterogeneous nucleation and alloying. *Int. Mater. Rev.* **47**, 3-29 (2002).

necessarily appear as “empty” cavities in the final microstructure.

**Response:** Thanks for your comments. We are sorry for our inaccurate statements and
we have corrected the relevant formulations as follows.

“As the solidification process advances, the proportion of liquid phase decreases. When
the tensile stress resulting from solidification shrinkage exceeds the strength of the
almost completely solidified microstructure and the liquid feeding is insufficient during
solidification, solidification cracking would occur between the dendrites²¹.”

8. The authors claim that the ZCASW weld did not contain any cracks. For
confirmation they present a crack-free 3D reconstruction captured using computed
tomography (CT). However, this confirmation is lacking for several reasons:

- The dimensions (L x B x H) of the cubic CT ‘microvolume’ as illustrated in Figure
1 are missing. If the dimensions of the CT volume are small in comparison to the
dimensions of the weld, existing (macro)cracks may possibly be overlooked.

- The exact position of the CT volume inside of the weld seam should be provided.
Did the authors extract the volume from the root, the top or the side section of the
weld, or did they even extract it from the heat affected zone (HAZ)? Stating that
“...a microvolume was excised from a region of [the] weld...” is not sufficient.

**Response:** Firstly, we observed the surface topography, longitudinal and cross section
morphology of the welding seam to detect the macro crack defects. If the crack existed,
we excised a microvolume with dimensions 2 mm×2 mm×5 mm from the region
containing crack defects. If the crack did not exist, we excised a microvolume with
dimensions 2 mm×2 mm×5 mm from the upper part of the center of the welding seam
for X-ray microtomography analysis. Fig. 1 shows the schematic diagram of the CT
samples preparation location. Fig. 2 shows the experimental result.

21. Kou, S. Welding metallurgy. USA: New Jersey (2003).

Fig. 1 The schematic diagram of the CT samples preparation location.

Fig. 2 3D-reconstruction volume of the defects inside the welding seam fabricated with the ZCASW filler (Color bar denotes pores diameter).

9. It is supposed to separate Figure 1 (a, b and c) that illustrates the filler wire and the process from Figure 1 (d and e) that shows the weld seam. Moreover, clear macroscopic images of the weld cross-section and of the weld surface are lacking and must be added to the manuscript! The authors did not present any high-resolution macroscopic image of the weld cross-section. Etching of the polished cross-section would be very beneficial to visualize the different microstructures of the weld, the HAZ and the base metal. Moreover, the cross-sections would enable to identify potential macrodefects, such as cracks or pores.

Response: Thank you very much for your insightful comments. We have separated Figure 1(a, b and c) from Figure 1. Meanwhile, we have added the clear macroscopic images of the welding seam to the revised manuscript, as shown in Fig. 3.

Fig. 3 The macroscopic morphology of the welding seam fabricated with the ZCASW filler material. a The surface topography of the welding seam. **b** The longitudinal section morphology of the welding seam. **c** The cross-sectional morphology of the welding seam.

10. The authors should be aware that Figure 2 (a, d, g) does not show any ‘grains’. The bright areas obviously represent the primary dendritic phase (aluminum dendrites), the dark areas represent the interdendritic phase. The EDX element distribution maps in Figure 2 (k) and in Supplementary Figure 5 do not confirm that Zr, Cu and Mg are present in the weld, because only black squares are shown!

Response: We have corrected the relevant expressions about the bright areas and the dark areas, and replaced them with the dendrites and the interdendritic phase. Meanwhile, we have added the chemical composition of the welding seam fabricated with the ZCASW filler to the revised manuscript, and the test was conducted using Inductive Coupled Plasma Optical Emission Spectrometer (ICP-OES), the results are shown in Table 1. The content of Zr, Cu and Mg is 1.8066%, 4.1957% and 1.4899%, respectively.

Table 1 Chemical composition of deposit fabricated with the ZCASW filler/wt%

Composition	Al	Cu	Mg	Ti	Mn	Zr
deposit	91.7117	4.1957	1.4899	0.0653	0.4934	1.8066

11. Figure 3 (a, c) is unnecessary, as it does not provide any additional information to Figure 3 (b, d). Figure 3 (e, f) should be larger to provide valuable information. The authors claimed that an eutectic formed in the channels between the dendrites. Is it really an eutectic (i.e., it formed by coupled solidification of two phases) or is it just the remaining melt that was highly enriched in Cu, Mg and other elements and that finally solidified as low-melting phase? Are the authors sure about the

550 crystallographic stoichiometry (S-Al₂CuMg)? Is there any evidence in the
551 corresponding ternary phase diagrams?

**Response:** Thank you for your valuable comment. We have corrected the original
Figure 3 and renamed it as Figure 4 in the revised manuscript, as shown in the following.
Meanwhile, we have modified the relevant expressions about the eutectic phase and
replaced them with the interdendritic phase. Additionally, we have recalibrated the
selected electron diffraction information for the interdendritic phase, and the result is
shown in Fig.4(f), which is consistent with Mg₂Cu phase. Fig.5 presents the Al-Mg-Cu
system phase diagram at 400 °C^{22,23}.

-
22. Chen, S. L. et al. A thermodynamic description for the ternary Al-Mg-Cu system. *Metall. Mater. Trans. A.* **28**, 435-446 (1997).
23. Raghavan, V. Al-Cu-Mg (Aluminum-Copper-Magnesium). *J. Phase Equilib. Diff.* **28**, 174-179 (2007).

**Fig. 4 Interdendritic phase morphology characterization in the melting zone fabricated with**

**the ER2319 and the ZCASW filler materials. a and b Interdendritic phase in the melting zone**

**fabricated with the ER2319 and the ZCASW filler materials. c Comparison of interdendritic phase**

**area fractions obtained from different melting zones. d EPMA maps showing the chemical**

**constitutions of the interdendritic phase and Al matrix in the melting zone fabricated with the**

**ZCASW filler. e Interdendritic phase observed by TEM-EDX, showing a STEM-HAADF image. f**

**SAED pattern of the interdendritic phase. g EDX mapping of the main elements (Al, Cu, Mg and**

**Zr). h HRTEM image of the interface between the α -Al and Mg_2Cu phase, with the inset showing**

**the FFT pattern of Mg_2Cu phase.**

Fig. 5 Al-Cu-Mg computed isothermal section at 400 °C^{22,23}

12. The authors should state where the values of the microstructural features that are
 summarized in Table 1 were taken from. Supplementary Table 1 should be part of
 the manuscript.

**Response:** Thanks for your comments. The grain size (d) was 4 μm derived from the
 EBSD analysis (Fig. 3h and i in the revised manuscript). The concentration of the solute
 element Mg and Cu (in wt.%) was measured by SEM-EDS (Supplementary Table 1).
 The dislocation densities (ρ_{GND}) was determined by EBSD analysis (Supplementary
 Fig.4). r and f were extracted from the SEM images in BSE mode (Supplementary Fig.6)
 using the image processing software. These characteristic values of the microstructure
 for strength calculation are mentioned in the estimating the strengthening contributions
 section of the methods in the revised manuscript. Meanwhile, we have added
 Supplementary Table 1 to the revised manuscript and renamed it as Table 2.

13. Supplementary Figure 7 can be removed, as it does not provide any valuable
 information. The results of the presented spot measurements (high aluminum peak,
 low Cu and Mg peaks) are trivial, as all spots were located at the primary aluminum

22. Chen, S. L. et al. A thermodynamic description for the ternary Al-Mg-Cu system. *Metall. Mater. Trans. A.* **28**, 435-446 (1997).
 23. Raghavan, V. Al-Cu-Mg (Aluminum-Copper-Magnesium). *J. Phase Equilib. Diff.* **28**, 174-179 (2007).

dendrites.

**Response:** We have removed the supplementary Fig.7.

Response to Referee #3:

**General Comments:**

This paper presents a new welding wire consisting of stranded aluminium wires around
a Zr wire at the core to weld ‘unweldable’ 2024 Al alloy. This is a subject particularly
interesting and relevant to the academic and industrial community. However, I am
raising a number of concerns that need to be addressed before this work can be accepted
for publication.

**Specific Comments:**

1. The new filler wire is proposed to avoid the formation of cracks in the weld but the
authors provide no evidence of the cracks present using traditional filler material.

**Response:** We performed laser welding experiments using ER2319 filler to fuse
AA2024 sheets. Meanwhile, to further visualize the 3D nature of internal imperfections,
a 2 mm×2 mm×5 mm microvolume was excised for examination by X-ray micro
computed tomography technology, the result is shown in Fig.6.

**Fig. 6 3D-reconstruction volumes of the defects inside the welding seam fabricated with the**
**ER2319 filler (Color bar denotes pores diameter).**

2. Fig. 1e shows a reconstructed 3D tomography image of the weld. The authors do
comment on the absence of crack but it does seem that a number of voids are present.
Can the authors comment on the importance of these voids? Are these comparable
to what would be obtained in a traditional fusion weld? Would this distribution of

voids be acceptable? It would have been interesting to perform a similar
tomography experiment with traditional fillers for comparison.

**Response:** Thank you for your suggestion. We conducted X-ray micro computed
tomography technology analysis on the welds fabricated with the ER2319, the ER4043
and the ZCASW fillers, respectively. The results are shown in Fig. 7a, b and c. The
porosity in the welds are 0.02%, 0.06% and 0.15%, respectively. Although the porosity
corresponding to the ZCASW filler is slightly higher than that corresponding to the
traditional fillers, it is commonly acceptable in engineering ($\leq 1\%$). How to further
reduce the porosity and improve the strength is also the focus of our further work.

**Fig. 7 3D-reconstruction volumes of the defects inside the welding seam fabricated with**
**different filler materials. a** The welding seam fabricated with the ER2319 filler. **b** The welding
seam fabricated with the ER4043 filler, and **c** The welding seam fabricated with the ZCASW filler
(Color bar denotes pores diameter).

3. Line 166 – The sentence does not really make sense, maybe ‘if’ needs to be removed.

**Response:** We have removed the ‘if’ from this sentence, as shown below.

“As the temperature and liquid volume fraction decrease, the long channels could be
trapped or hindered by the developing dendritic solid network.”

4. The authors mention the formation of cracks when using alternative fillers,
particularly the ER2319 filler. However, no example of such cracks are shown. It
would be beneficial to add such images of cracks to validate these claims.

**Response:** We have added the 3D-reconstruction volumes image of the defects inside
the welding seam fabricated with the ER2319 filler to the revised manuscript, as shown
in Fig.6.

**Fig.6** 3D-reconstruction volumes of the defects inside the welding seam fabricated with the
ER2319 filler (Color bar denotes pores diameter).

5. Line 247 and 256. The higher magnification images are Fig 3b and Fig 3d (and not
3a and 3c)

**Response:** We have corrected the original Fig. 3 and renamed it as Fig. 4 in the revised
manuscript. To observe the interdendritic phase morphology more clearly and
intuitively, we just retained the same magnification images of the melting zone
fabricated with the ER2319 and the ZCASW fillers in the backscattered electron (BSE)
diffraction mode.

6. The grain sizes from the EBSD in Fig 2 do not seem to match the ones from the
microscopy presented in Fig 3. In Fig 3b and 3d, the grain size appears similar for
the ER2319 and ZCASW.

**Response:** We have regained the image of the melting zone fabricated with the ER2319
and the ZCASW fillers in the backscattered electron (BSE) diffraction mode at the same

magnification, as shown in Fig. 8.

**Fig. 8 Interdendritic phase morphology characterization in the melting zone fabricated with**
**the ER2319 and the ZCASW filler materials. a** Interdendritic phase in the melting zone fabricated
with the ER2319 filler. **b** The interdendritic phase in the melting zone fabricated with the ZCASW
filler.

7. It seems difficult to conclude on the morphology of the eutectic phases between fig
3b and 3d. It seems that a large amount of the eutectic phases has been dislodged
during the polishing step leaving a high number of voids at the GBs. So it seems
difficult to conclude that ‘the occurrence and size of the eutectic features were
drastically reduced’ from these images only.

**Response:** We are very sorry for our inaccurate statements about eutectic phase and
have replaced them with the interdendritic phase. Meanwhile, we have corrected the
relevant formulation related to the interdendritic phase morphology of the melting zone
fabricated with the ER2319 and the ZCASW fillers in the revised manuscript, as shown
below.

“Fig. 4a illustrates the typical interdendritic phase distribution in the melting zone
fabricated with the ER2319 filler. The interdendritic phase is lamellar and continuously
distributes along the interdendritic regions, normally resulting in continuous
segregations.”

“Fig. 4b displays the typical interdendritic phase distribution in the melting zone
fabricated with the ZCASW filler. Here, the interdendritic phase is segmented, and its
fragments are randomly oriented, shorter than their counterparts fabricated with the
ER2319 filler.”

8. Again, it is difficult to conclude anything from the CT provided in 3e as there is no

discussion on the size and volume fraction of the eutectic features and crucially no
comparison with the weld obtained with the alternative filler material.

**Response:** We have corrected the relevant formulation related to the eutectic phase and
replaced them with interdendritic phase. Meanwhile, we discussed the interdendritic
phase characteristics based on Fig. 9a and b, and compared the area fraction of the
interdendritic phase (The bright areas) based on Fig. 9c and d (The binarized image of
Fig. 9a and b). The results are shown in Fig. 9e, the area fraction of the interdendritic
phase in the melting zone fabricated with the ER2319 filler is 11.5%, and it increases
to 15.95% in the melting zone fabricated with ZCASW filler. More interdendritic phase
indicates more liquid exists among dendrites during the terminal stage of solidification⁸,
which could sufficiently compensate for solidification shrinkage and thermal
contraction.

8. Zhao, Y. N. et al. New alloy design approach to inhibiting hot cracking in laser additive manufactured nickel-based superalloys. *Acta Mater.* **247**, 118736 (2023).

**Fig. 9 Interdendritic phase morphology characterization in the melting zone fabricated with**
 **the ER2319 and the ZCASW filler materials. a and b** The interdendritic phase in the melting
 zone fabricated with the ER2319 and the ZCASW filler materials. **c and d** The binarized image of
 **a and b. e** Comparison of interdendritic phase area fractions obtained from different melting zones.

9. The solidification behaviour is assessed via simulating the evolution of the liquid
 channel morphology between the two different fillers. A cracking susceptibility
 index is calculated as the slope of the T vs $f_s^{0.5}$ plot. The CSI is said to be higher
 for the ER2319 vs ZCASW wire. However, the curves are so close to each other
 that the difference between the two seems almost negligible. How relevant is the
 observed difference in CSI. Also, could you plot the derivative of T vs $f_s^{0.5}$ which
 would make it easier to conclude on whether there is a clear difference. At the

moment it seems difficult to conclude from the information provided.

**Response:** Thank you for your valuable comments. We have plotted the curve of
$|dT/d(f_s^{1/2})|$ vs T at the final solidification stage in the inset of Fig. 10. It is evident that
the cracking susceptibility index corresponding to the ER2319 material is higher than
that corresponding to the ZCASW filler.

**Fig.10** T - $(f_s)^{1/2}$ curves and cracking susceptibility indexes calculated from the PF simulations
for the melting zone fabricated with the ZCASW (blue) and the ER2319 (orange) fillers.

10. The mechanical properties of the different joints are then analysed using a
combination of hardness and tensile tests. The way the tensile strength is reported
is confusing, do you quote the highest obtained strength and then in brackets the
average and standard deviation? It would be more standard to only report the
average value. How many tensile tests were conducted per condition?

**Response:** Thank you for your advice. In the previous manuscript, we reported the
highest tensile strength and then listed the average and standard deviation in brackets.
In the revised manuscript, we only reported the average value of tensile strength and
conducted three experiments under per condition.

11. When assessing the different contributions to strengthening, how were the
precipitates analysed and what precipitates are we talking about? Is that the coarse
Al₃Zr phases?

**Response:** The precipitates analyzed when assessing the different contributions to
strengthening is the coarse Al₃Zr phases. We obtained the mean precipitate radius r and
the precipitate volume fraction f of the Al₃Zr phases from the SEM images in the BSE
diffraction mode (supplementary Fig. 6) using the image processing software. Then, we
assessed the strength contribution arising from the Orowan bypassing mechanism using
the Eq. (5) in revised manuscript.

REVIEWERS' COMMENTS

Reviewer #1 (Remarks to the Author):

All technical concerns of the reviewer have been sufficiently addressed.

Reviewer #2 (Remarks to the Author):

The reviewer acknowledges that the authors tried to comprehensively address the comments of the first review round.

Reviewer #3 (Remarks to the Author):

The Authors have addressed all of the reviewers' comments and concerns